# Selective Expert Guidance For Effective and Diverse Exploration in Reinforcement Learning of LLMs

**Zishang Jiang**[1], **Jinyi Han**[2], **Tingyun Li**[1], **Xinyi Wang**[1], **Sihang Jiang**[3], **Zhaoqian Dai**[4],
**Shuguang Ma**[4], **Fei Yu**[4], **Jiaqing Liang**[1]*, **Yanghua Xiao**[3]

[1]School of Data Science, Fudan University
[2]Shanghai Institute of Artificial Intelligence for Education, East China Normal University
[3]College of Computer Science and Artificial Intelligence, Fudan University
[4]Ant Group
{zsjiang24, xinywang24}@m.fudan.edu.cn,
{sihangjiang, liangjiaqing, shawyh}@fudan.edu.cn
{jinyihan099, litinyun0715, feiyu.fyyu}@gmail.com,
{daizhaoqian.dzq, liangxiao.msg}@antgroup.com

## Abstract

Reinforcement Learning with Verifiable Rewards (RLVR) has become a widely adopted technique for enhancing the reasoning ability of Large Language Models (LLMs). However, the effectiveness of RLVR strongly depends on the capability of base models. This issue arises because it requires the model to have sufficient capability to perform high-quality exploration, which involves both effectiveness and diversity. Unfortunately, existing methods address this issue by imitating expert trajectories, which improve effectiveness but neglect diversity. To address this, we argue that the expert only needs to **provide guidance at critical decision points** rather than the entire reasoning path. Based on this insight, we propose **MENTOR**: Mixed-policy Expert Navigation for Token-level Optimization of Reasoning, a framework that provides expert guidance only at critical decision points to perform effective and diverse exploration in RLVR. Extensive experiments show that MENTOR enables models capture the essence of expert strategies rather than surface imitation, thereby performing high-quality exploration and achieving superior overall performance. Our code is available online[1].

## 1 Introduction

Reinforcement Learning with Verifiable Rewards (RLVR) has become a widely adopted technique for enhancing the reasoning ability of Large Language Models (LLMs). It has significantly improved models' performance in solving challenging mathematics and programming problems, as evidenced by models such as OpenAI-o1 (Jaech et al., 2024), DeepSeek-R1 (Guo et al., 2025), and Kimi-1.5 (Team et al., 2025). These improvements are largely attributed to the models' ability to generate detailed chains of thought (CoT) before giving final answers (Wei et al., 2022), which is termed test-time scaling (Muennighoff et al., 2025).

However, the effectiveness of RLVR strongly depends on the capability of base models. It has been observed that when applied to models with limited parameters, RLVR fails to reproduce the remarkable gains observed on powerful base models (Guo et al., 2025).

This issue arises because RLVR requires the model to have sufficient capability to perform high-quality exploration, which involves both **effectiveness** and **diversity**. Specifically, when the task is overly challenging for the model, it often struggle to discover any correct reasoning trajectory (Yue et al., 2025), resulting in ineffective exploration that hinders training (Yu et al., 2025). Furthermore,

---

*Corresponding author.
[1]https://github.com/Jiangzs1028/MENTOR

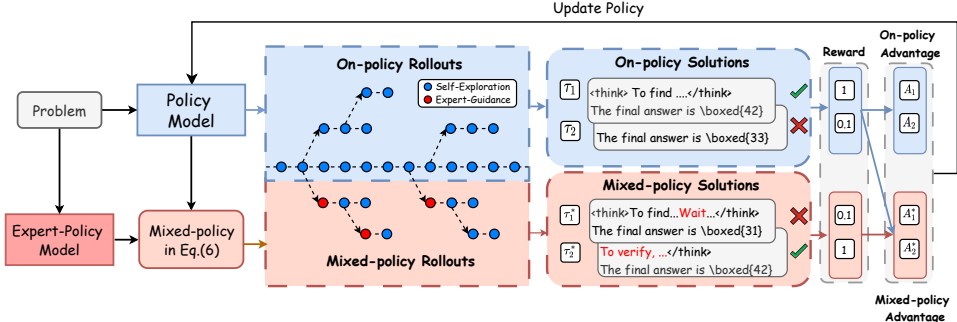

Figure 1: Illustration of MENTOR framework. By providing expert guidance only at critical decision points, MENTOR steers reasoning trajectories while preserving the policy's own exploration, thereby avoiding the constraints of fixed expert trajectories and achieving more effective and diverse exploration in RL training.

even when correct solutions are found, limited diversity of reasoning trajectories often leads the model to rapidly converge to a narrow set of solutions (Song et al., 2025), which reflected in entropy collapse (Cui et al., 2025) and ultimately traps it in suboptimal solutions (Song et al., 2025).

Unfortunately, existing methods address this issue by imitating expert trajectories, which **improve effectiveness but neglect diversity**. While such imitation reduces ineffective exploration (Yan et al., 2025; Zhang et al., 2025a;b; Liu et al., 2025; Li et al., 2025), it forces the model to follow to fixed expert trajectories, thereby restricting the diversity of exploration and accelerating entropy collapse (Yan et al., 2025). In addition, the reduction of diversity is further accelerated by gradient imbalance (Huang et al., 2025), which drives the model to quickly overfit expert trajectories, especially when expert reasoning patterns diverge substantially from those of the policy model (Zhang et al., 2025a). Although some works attempt to mitigate it by reweighting tokens in expert trajectories (Yan et al., 2025; Zhang et al., 2025a), the relief remains superficial, as the exploration space is still fundamentally restricted by the fixed expert trajectories.

To achieve better exploration, we argue that the expert only needs to **provide guidance at critical decision points** rather than the entire reasoning trajectory. Expert guidance is indeed essential for steering the model toward correct solutions, but blindly imitating full expert trajectories restricts the exploration space. Since tokens contribute unequally to reasoning trajectories (Wang et al., 2025), introducing guidance at critical decision points enables the model to best leverage expert knowledge while preserving exploration diversity. Based on this insight, we propose **MENTOR**: Mixed-policy Expert Navigation for Token-level Optimization of Reasoning, a framework that injects expert guidance only at critical decision points to perform effective and diverse exploration. Extensive experiments show that MENTOR enables models capture the essence of expert strategies rather than surface imitation, thereby sustaining high-quality exploration and achieving superior overall performance.

Our contributions can be summarized as follows:

- We provide a formal analysis of RLVR and demonstrate that effective policy improvement critically depends on high-quality exploration, which requires not only discovering correct solutions but also maintaining sufficient diversity to prevent entropy collapse and avoid being trapped in suboptimal solutions.

- We are the first to propose leveraging expert knowledge only at critical decision points in RLVR training rather than imitating entire expert trajectories, thereby enabling models to achieve both effective and diverse exploration in RLVR.

- We conduct extensive experiments showing that MENTOR delivers consistent improvements on six challenging math benchmarks and out-of-domain tasks, with gains stable across diverse model families. Further analysis reveals that it mitigates entropy collapse in RLVR training and broadens the capability boundary of base models, and case studies demonstrate it can selectively absorb expert knowledge rather than superficial imitation.

## 2 WHAT IS HIGH-QUALITY EXPLORATION IN RLVR?

Exploration is fundamental to reinforcement learning, as it enables models to discover more rewarding strategies and thereby avoid being trapped in suboptimal behaviors. In this section, we investigate the necessary conditions of high-quality exploration in RLVR.

### 2.1 PRELIMINARY

Let $\mathcal{S}$ denote the space of all possible token sequences over the LLM's vocabulary, and let $\pi_\theta$ denote a LLM with parameters $\theta$. Given a question space $\mathcal{D} \subseteq \mathcal{S}$ and a input $q \in \mathcal{D}$, the model generates sequences $\tau$ autoregressively according to a conditional distribution $\pi_\theta(\cdot|q)$.

**Definition 2.1** (Exploration Support Set). Given a probability threshold $\delta_p$ and a question $q$, define the exploration support of $\pi_\theta(\cdot|q)$ that excludes negligible-probability sequences:

$$\text{supp}(\pi_\theta(\cdot|q)) = \left\{ \tau \in \mathcal{S} \big| \pi_\theta(\tau|q) > \delta_p \right\}, \tag{1}$$

Although softmax guarantees that every sequence has strictly positive probability, a limited sampling budget makes extremely low-probability sequences practically unreachable. Therefore, $\text{supp}(\pi_\theta(\cdot|q))$ characterizes the effective exploration space of the model for a given question $q$.

Fine-tuning LLM $\pi_\theta$ using RL with a reward function $R(\cdot)$ involves repeatedly sampling sequences from the current policy, rewarding the LLM for correct sequences and penalizing for the wrong ones, in order to maximize the expected reward:

$$J(\theta) = \mathbb{E}_{q \sim \mathcal{D}, \tau \sim \pi_\theta(\cdot|q)}[R(q, \tau)]. \tag{2}$$

In practice, this objective is commonly optimized with **Group Relative Policy Optimization (GRPO)** (Shao et al., 2024), which has demonstrated strong performance across tasks and enables effective scaling in the RLVR paradigm. GRPO leverages the reward scores of $G$ sampled solutions for a given question $q$ to estimate advantages, thereby eliminating the need for an additional value model. Formally, let $\pi_{\theta_{\text{old}}}$ and $\pi_\theta$ denote the policy before and after the update, each representing a distribution over tokens at every position. Given a question $q$, a set of sampled solution sequences $\tau_i$ from $\pi_{\theta_{\text{old}}}$, and a reward function $R(\cdot)$, GRPO computes the advantage $A_i$ by normalizing rewards within the group,

$$\mathcal{J}_{\text{GRPO}}(\theta) = \mathbb{E}_{q \sim \mathcal{D}, \{\tau_i\}_{i=1}^G \sim \pi_{\theta_{\text{old}}}(\cdot|q)}$$

$$\left[ \frac{1}{\sum_{i=1}^G |\tau_i|} \sum_{i=1}^G \sum_{t=1}^{|\tau_i|} \min\left( r_{i,t}(\theta)\hat{A}_{i,t}, \text{clip}\left(r_{i,t}(\theta), 1 \pm \varepsilon_{\text{clip}}\right)\hat{A}_{i,t}\right) - \beta D_{\text{KL}}(\pi_\theta||\pi_{\text{ref}})\right] \tag{3}$$

where

$$r_{i,t}(\theta) = \frac{\pi_\theta\left(\tau_{i,t} \mid q, \tau_{i,<t}\right)}{\pi_{\theta_{\text{old}}}\left(\tau_{i,t} \mid q, \tau_{i,<t}\right)}, \quad \hat{A}_{i,t} = \frac{R_i - \text{mean}(\{R_i\}_{i=1}^G)}{\text{std}(\{R_i\}_{i=1}^G)}. \tag{4}$$

### 2.2 THE NECESSARY CONDITIONS OF HIGH-QUALITY EXPLORATION IN RLVR

**Definition 2.2** (Explorable Optimal Trajectory Subset). For a given question $q$, the optimal trajectory set within the exploration support $\text{supp}(\pi_\theta(\cdot|q))$ is defined as

$$\mathcal{T}^\star = \{\tau \in \text{supp}(\pi_\theta(\cdot|q)) \mid R(q, \tau) = R_{\max}(q)\}, \tag{5}$$

where $R_{\max}(q) = \sup_{\tau \in \mathcal{S}} R(q, \tau)$ denotes the maximal achievable reward for question $q$ .

Intuitively, $\mathcal{T}^\star$ is a subset of the globally optimal trajectories, representing the portion of optimal solutions that the model can actually sample during rollouts. Under the training objective in Eq. (2), the support of $\pi_\theta$ will progressively contract toward $\mathcal{T}^\star$, eventually concentrating its probability mass on this set. This convergence yields the optimal policy $\pi_{\text{opt}}$, which maximizes the expected reward while maintaining the highest possible output diversity (see Appendix A.1 for proof).

**Effectiveness issue.**    However, a key insight is that if the model lacks the ability to discover any optimal trajectory, then $\mathcal{T}^\star$ becomes empty, and the reinforcement learning process can no longer make progress. For example, under GRPO, when correct solutions are absent, the normalized advantages $\hat{A}_{i,t}$ in Eq. (4) tend to approach zero. Consequently, the update term in Eq. (3) becomes ineffective, preventing any policy improvement. Therefore, a necessary condition for high-quality exploration is that the policy must be able to discover at least one optimal trajectory within its support.

**Diversity issue.**    During reinforcement learning, policy entropy tend to rapidly collapse, leading to reduced diversity in model outputs and limiting the exploration of a wider range of possible trajectories. Some studies have found that the decline in exploratory diversity can hinder performance improvements on unsolved problems (Song et al., 2025). The following theorem formalizes this diversity issue (a detailed proof is provided in the appendix A.1):

**Theorem 2.1** (Entropy Upper-Bound Decay with Increasing Expected Reward).  In the binary-reward case $R \in \{0, 1\}$, let $\mathcal{T}^\star$ be the set of optimal trajectories with $K = |\mathcal{T}^\star|$, $M = |\mathcal{S}_q \setminus \mathcal{T}^\star|$, $N = K + M$, where $|\cdot|$ denotes the cardinality of a set. For any expected reward $C \in (0, 1)$, the policy entropy $H$ is upper-bounded by $H_{\mathrm{ub}}(C)$, given by

$$H \leq H_{\mathrm{ub}}(C) = H_{\mathrm{b}}(C) + C \log K + (1 - C) \log M. \tag{6}$$

where $H_{\mathrm{b}}(C) = -C \log C - (1 - C) \log(1 - C)$. For $c_2 > c_1$ with $c_1$ larger than the expected reward under the uniform policy on $\mathrm{supp}(\pi_\theta(\cdot \mid q))$ (i.e., $c_1 > \frac{K}{N}$), the entropy upper bound satisfies the single inequality

$$0 \ < \ H_{\mathrm{ub}}(c_1) - H_{\mathrm{ub}}(c_2) = (c_2 - c_1) \log \frac{N}{K} + H_b(c_1) - H_b(c_2), \tag{7}$$

The entropy upper bound necessarily decreases as the expected reward increases, with the amount of inversely proportional to the size $K$ of the optimal trajectory set $\mathcal{T}^\star$.

This theorem shows that to prevent a rapid collapse of diversity, high-quality exploration must ensure the discovery of multiple, diverse optimal trajectories. When the set $\mathcal{T}^\star$ contains only a few optimal solutions, increasing expected reward necessarily forces the policy to concentrate probability mass more aggressively, causing its entropy upper bound to drop rapidly and thus accelerating diversity collapse. In contrast, a larger $\mathcal{T}^\star$ can slow down entropy collapse and thus preserve more exploration diversity, thereby enabling the policy ultimately achieve higher final performance. Therefore, another necessary condition for high-quality exploration is that the policy must discover multiple distinct optimal trajectories, so that exploration diversity can be preserved during reward improvement.

> **Highlights**
>
> In summary, to avoid suboptimal convergence under limited exploration budgets, high-quality exploration is indispensable. Specifically, it must satisfy two necessary conditions: **effectiveness** and **diversity**. If either of these conditions is missing, the model will converge to a suboptimal solution.

## 3    MENTOR: MIXED-POLICY EXPERT NAVIGATION FOR TOKEN-LEVEL OPTIMIZATION OF REASONING

As discussed in Section 2, high-quality exploration in RLVR requires both effectiveness and diversity. However, existing methods that incorporate expert solutions improve effectiveness but overlook diversity, leading to entropy collapse (Zhang et al., 2025a). To address this, we propose **MENTOR**, a framework that balances effectiveness and diversity through two components: **Mixed-policy Rollout**, which introduces expert guidance only at critical decision points, and **Mixed-policy GRPO**, which integrates these guided rollouts into on-policy RL with modified advantage estimation. The overall framework is illustrated in Figure 1.

## 3.1 Mixed-policy Rollout

Existing expert-guided methods, in order to obtain reasoning trajectories beyond the capability of the base model, typically sample full trajectories from the expert model $\pi^*$, where every token is generated according to $y_t \sim \pi^*(\cdot \mid q, y_{<t})$, and the base model is then trained to imitate each token in this expert-generated trajectory equally.

However, recent studies show that tokens contribute unequally to reasoning trajectories (Wang et al., 2025). some (e.g., high-entropy tokens) determine critical decision forks, while others only serve as deterministic following. The latter often vary across models in stylistic ways, but such differences have little impact on reasoning process. Entire expert trajectories inevitably contain many of these low-impact tokens, which distract the model from learning the key reasoning decisions. To mitigate this problem, we introduce expert guidance only where it is truly needed.

**Definition 3.1** (Mixed-policy Distribution) At each decoding step $t$, we define a token-level mixed-policy distribution that interpolates between the on-policy distribution $\pi_\theta$ and the expert distribution $\pi^*$. The expert distribution $\pi^*$ is derived from a stronger reference model with the same vocabulary $\mathcal{V}$, such as a larger model or a domain-adapted model (Du et al., 2024). Formally, given question $q$ and prefix $y_{<t}$, the sampling distribution for token $y_t$ is:

$$\pi_{\text{mix}}(\cdot \mid q, y_{<t}) = (1 - w_t)\,\pi_\theta(\cdot \mid q, y_{<t}) + w_t\,\pi^*(\cdot \mid q, y_{<t}), \tag{8}$$

where $w_t = \min(1, H_t/\gamma_p)$ is the interpolation weight determined by the token-level entropy $H_t = -\sum_y \pi_\theta(y \mid q, y_{<t}) \log \pi_\theta(y \mid q, y_{<t})$, and $\gamma_p$ denotes the $p$-quantile of entropies across tokens in the batch. Thus, high-entropy tokens receive stronger expert guidance, while low-entropy tokens remain closer to the on-policy distribution $\pi_\theta$.

By sampling trajectories from this mixed-policy distribution, exploration achieves a balance between effectiveness and diversity. Effectiveness is enhanced because expert guidance is injected at uncertain decision points, increasing the probability of discovering correct trajectories. Diversity is preserved because expert guidance is restricted to only a few positions, ensuring that the exploration space remains exponentially large and avoiding collapse to a fixed expert solution. At the same time, selective guidance enables models to focus on learning the core reasoning strategies from the expert.

**Accelerating Mixed-policy Rollout.** Although $\pi_{\text{mix}}$ introduces expert guidance only at critical tokens, standard auto-regressive sampling from $\pi_{\text{mix}}$ still requires forward computation of both the policy model $\pi_\theta$ and the expert $\pi^*$ at every step to determine whether guidance is required, which substantially increases rollout cost and consequently reduces the efficiency of training, especially when the expert has a large number of parameters.

Since $\pi_{\text{mix}}$ deviates from the policy distribution $\pi_\theta$ only on a few tokens, while at the remaining positions $\pi_{\text{mix}}$ is close to $\pi_\theta$. Based on this positional sparsity, we propose an accelerated mixed-policy rollout method based on Speculative Sampling (Chen et al., 2023). Speculative Sampling is an unbiased acceleration method that let the draft model propose multiple tokens and then verifying them with the target model in parallel. Its acceleration effect depends on the draft acceptance rate, making it naturally suitable for mixed-policy rollout where most tokens align with the policy distribution.

We first let the policy model $\pi_\theta$ auto-regressively generate $K$ candidate tokens $\tilde{y}_{1:K}$, while recording the corresponding sampling distributions $\pi_\theta(\cdot|q, \tilde{y}_{<t})$ at each step $t$. Next, the expert model computes the distributions $\pi^*(\cdot|q, \tilde{y}_{<t})$ in parallel. Based on these results, we construct the mixed-policy distribution $\pi_{\text{mix}}(\cdot|q, \tilde{y}_{<t})$ as defined in Eq.(8). Each candidate token $\tilde{y}_t$ is then validated with the acceptance probability

$$\min\left(1, \frac{\pi_{\text{mix}}(\tilde{y}_t \mid q, \tilde{y}_{<t})}{\pi_\theta(\tilde{y}_t \mid q, \tilde{y}_{<t})}\right). \tag{9}$$

If $\tilde{y}_t$ is accepted, the process continues to the next candidate until either a rejection occurs or all $K$ candidates are accepted.

When a candidate is rejected, it is resampled from the residual distribution

$$\left(\pi_{\text{mix}}(\cdot \mid q, \tilde{y}_{<t}) - \pi_\theta(\cdot \mid q, \tilde{y}_{<t})\right)_+. \tag{10}$$

where $(f(v))_+ = \max(0, f(v)) / \sum_v \max(0, f(v)), \quad v \in \mathcal{V}$.

This process is repeated to generate complete sequences, enabling substantially faster sampling from the mixed policy while remaining unbiased with Eq.(8), see Appendix A.2 for proof. The detailed algorithm is summarized in Algorithm 1.

---

**Algorithm 1** Accelerating Mixed-policy Rollout with Modified Speculative Sampling

---

Given lookahead $K$, entropy threshold $\gamma_p$ and maximum response length $T$.
Given expert model $\pi^*$, and on-policy model $\pi_\theta$, question sequence $q$.
Initialize $n = 0$.
**while** $n < T$ **do**
  **for** $t = 1 : K$ **do**
    Sample candidate tokens from the policy model $\tilde{y}_t \sim \pi_\theta(\cdot|q, y_{\leq n}, \tilde{y}_{<t})$
    Compute the token-level entropy $H_t$ from the on-policy distribution $\pi_\theta(\cdot|q, y_{\leq n}, \tilde{y}_{<t})$
    Compute weight $w_t \leftarrow \min(1, H_t/\gamma_p)$
  **end for**
  In parallel, compute $K$ sets of logits from candidate tokens $\tilde{y}_1, \ldots, \tilde{y}_K$ :
$$\pi^*(\cdot|q, y_{\leq n}), \ \pi^*(\cdot|q, y_{\leq n}, \tilde{y}_1), \ldots, \ \pi^*(\cdot|q, y_{\leq n}, \tilde{y}_{<K})$$
  **for** $t = 1 : K$ **do**
    Sample $r \sim U[0, 1]$ from a uniform distribution.
    Compute $\pi_{\text{mix}}(\cdot|q, y_{\leq n}) \leftarrow (1 - w_t)\pi_\theta(\cdot|q, y_{\leq n}) + w_t\pi^*(\cdot|q, y_{\leq n})$
    **if** $r < \min\left(1, \frac{\pi_{\text{mix}}(\tilde{y}_t|q, y_{\leq n})}{\pi_\theta(\tilde{y}_t|q, y_{\leq n})}\right)$, **then**
      Set $y_{n+1} \leftarrow \tilde{y}_t$ and $n \leftarrow n + 1$.
    **else**
      sample $y_{n+1} \sim (\pi_{\text{mix}}(\cdot|q, y_{\leq n}) - \pi_\theta(\cdot|q, y_{\leq n}))_+$ and exit for loop.
    **end if**
  **end for**
**end while**

---

## 3.2 MIXED-POLICY GRPO

To effectively integrate samples generated by the mixed-policy rollout into GRPO, we extend the algorithm with a modified advantage function. Specifically, for each query $q$, we collect two sets of trajectories: (i) on-policy rollouts $\mathcal{G}_{on} = \{\tau\}^{N_1}$ sampled from the policy model $\pi_\theta$, and (ii) mixed-policy rollouts $\mathcal{G}_{\text{mix}} = \{\tau\}^{N_2}$ sampled from the mixed-policy $\pi_{\text{mix}}$. Then optimizes the policy model by maximizing the following objective:

$$\mathcal{J}_{\text{mixed}}(\theta) = \frac{1}{\sum_{i=1}^{N_1+N_2}|\tau_i|} \sum_{i=1}^{N_1+N_2} \sum_{t=1}^{|\tau_i|} \min\left(r_{i,t}(\theta)\hat{A}_{i,t}, \text{clip}\left(r_{i,t}(\theta), 1 - \varepsilon, 1 + \varepsilon\right)\hat{A}_{i,t}\right) \quad (11)$$

**On-policy advantages.** For $\tau \in \mathcal{G}_{on}$, we retain GRPO's group-wise standardization to promote self-improvement:

$$\hat{A}_{i,t}(\tau) = \frac{R_i - \text{mean}\left(\{R_j\}_{\tau_j \in \mathcal{G}_{on}}\right)}{\text{std}\left(\{R_j\}_{\tau_j \in \mathcal{G}_{on}}\right)}, \quad \tau \in \mathcal{G}_{on}. \quad (12)$$

**Mixed-policy advantages.** For $\tau \in \mathcal{G}_{\text{mix}}$, we aim to **encourage exploration rather than penalize failures**. To this end, we define its advantage function as the positive excess of its reward over the mean reward of on-policy rollouts:

$$\hat{A}_{i,t}(\tau) = \alpha \cdot \frac{\left[R_i - \text{mean}\left(\{R_j\}_{\tau_j \in \mathcal{G}_{on}}\right)\right]_+}{R_{\text{range}}}, \quad \tau \in \mathcal{G}_{\text{mix}}. \quad (13)$$

where $[x]+ = \max(x, 0)$ ensures that only above-average exploration is rewarded while failures are ignored, and $R_{\text{range}}$ is a fixed reward span (e.g., the global maximum–minimum reward range) used to normalize rewards into $[0, 1]$ for numerical stability. And $\alpha$ is a weighting coefficient that balances the contribution of samples from the mixed-policy. In our setting, $\alpha$ is additionally scheduled to gradually decay, thereby shifting the policy from expert-guided exploration to self-driven exploration as training progresses.

## 4 EXPERIMENTS

### 4.1 SETUP

**Datasets and Models.** We conduct experiments on two model families: Qwen2.5 (Team, 2024) and LLaMA3.1 (Dubey et al., 2024). For Qwen2.5, we use the Qwen2.5-7B-Base and Qwen2.5-3B-Base for experiments. And we use the MATH dataset (Hendrycks et al.) as training dataset, restricting to problems with difficulty levels 3–5 and removing any instances overlapping with the test set to prevent data leakage, total 8,889 training examples. For LLaMA3.1, we use the LLaMA3.1-8B-Base for experiments. However, the MATH dataset is too difficult for this model, such that vanilla GRPO fails to train successfully. To enable comparison between GRPO and other baselines, we construct a simplified dataset from OpenR1-MATH-220K[2] (Hugging Face, 2025) as the training dataset for LLaMA3.1. Further dataset and expert model details are provided in the Appendix C.

**Evaluations.** We evaluate the models along two categories. (i) **In-domain performance.** We assess the in-domain performance on mathematics benchmarks, including MATH (Hendrycks et al.), AIME24, AIME25, and AMC (Li et al., 2024). (ii) **Out-of-domain performance.** To examine whether post-tuning affects general reasoning ability beyond mathematics, we further evaluate the out-of-domain performance in MMLU-Pro (Wang et al., 2024) and GPQA-diamond (Rein et al.). For AIME24, AIME25, and AMC, we report avg@32 at temperature 0.6 as the test set is relatively small, while for the other benchmarks, we report pass@1 at temperature 0.

**Baselines.** We compare MENTOR with several representative baselines, including: (1) **Base**: The base model without any fine-tuning. (2) **On-policy RL**: Standard GRPO without expert guidance, enhanced with token-level loss and the Clip-Higher in DAPO (Yu et al., 2025) to serve as a stronger baseline. (3) **LUFFY** (Yan et al., 2025): A method that integrates full expert trajectories within the GRPO rollout groups. (4) **QuestA** (Li et al., 2025): A method that provides the first half of expert trajectories as hints for the model to follow. Hyper-parameters and training details of different methods can be found in Appendix C.

### 4.2 MAIN RESULTS

**MENTOR achieves consistent improvements across different models.** Table 1 shows that MENTOR outperforms the on-policy RL baseline across all three backbones. On Qwen2.5-7B, for example, MENTOR lifts the average score on the MATH benchmark from 76.8 to 81.4, and yields notable relative gains of +4.1, +7.4, and +7.1 points on AIME24, AIME25, and AMC, respectively. Similar trends are observed on Qwen2.5-3B and LLaMa3.1-8B. Importantly, these gains are not confined to in-domain reasoning. MENTOR also delivers clear improvements on out-of-domain benchmarks, demonstrating that the reasoning abilities learned under expert guidance can effectively generalize to out-of-domain tasks.

**MENTOR achieves a better trade-off between expert guidance and autonomous exploration.** Compared to on-policy RL, LUFFY introduces full expert trajectories but achieves only limited improvements across all models, indicating that directly imitating expert solutions does not fully leverage expert knowledge. This is likely because full trajectories overly constrain the exploration space, causing the model to overfit superficial expert patterns and fall into suboptimal strategies. QuestA, which provides partial expert trajectories as hints, alleviates over-imitation to some extent but its effectiveness strongly depends on model capacity: it yields clear gains (+1.3) on Qwen2.5-7B, only minor improvement (+0.8) on Qwen2.5-3B, and even a negative effect (-1.6) on LLaMa3.1-8B. This is because, in the absence of subsequent guidance, the weaker model struggles to explore correct solutions, and the excessive hints further disrupt its exploration. In contrast, MENTOR consistently outperforms across different models, achieving a better balance between leveraging expert knowledge and maintaining autonomous exploration, thereby achieving significant improvements.

Table 1: MENTOR vs. other baselines. **Compared to the On-policy RL, MENTOR achieves an average performance improvement of 3.2%, 4.3% and 3.9% on the three models, respectively.** The best results are highlighted in bold, and the second-best results are underlined.

| Methods | In-Domain Performance | | | | | | Out-of-Domain | | | Avg |
|---|---|---|---|---|---|---|---|---|---|---|
| | MATH | AIME24 | AIME25 | AMC | Minerva | Olympiad | GPQA | ARC | MMLU-Pro | |
| LLaMa3.1-8B-Base | | | | | | | | | | |
| Base | 10.6 | 0.1 | 0.0 | 1.8 | 4.4 | 2.1 | 0.0 | 0.0 | 0.1 | 2.1 |
| On-policy RL | 24.0 | 0.4 | 0.4 | 8.0 | 13.6 | 6.4 | 25.8 | 70.7 | 35.7 | 20.6 |
| LUFFY | 25.2 | **0.5** | 0.4 | 8.4 | 14.0 | 7.1 | 27.8 | 74.9 | 34.9 | 21.5 |
| QuestA | 20.6 | 0.1 | 0.2 | 5.3 | 8.8 | 4.0 | 25.3 | 72.5 | 33.9 | 19.0 |
| MENTOR | **30.2** | **1.2** | **0.6** | **10.4** | **16.2** | **8.9** | **30.3** | **77.3** | **39.1** | **23.8** |
| Qwen2.5-3B-Base | | | | | | | | | | |
| Base | 47.4 | 2.4 | 1.9 | 17.7 | 19.9 | 19.0 | 3.0 | 23.6 | 19.4 | 17.1 |
| On-policy RL | 65.8 | 3.3 | 2.5 | 32.2 | 25.4 | 29.8 | 17.7 | 72.1 | 30.6 | 31.0 |
| LUFFY | 64.0 | 5.2 | **4.2** | 32.8 | 25.0 | 30.1 | 15.2 | 72.5 | 30.8 | 31.1 |
| QuestA | 66.4 | 7.9 | 2.9 | 34.1 | **27.6** | 29.8 | 16.2 | 70.3 | 30.9 | 31.8 |
| MENTOR | **69.8** | **8.3** | 3.8 | **34.2** | 26.5 | **35.2** | **22.7** | **80.8** | **36.8** | **35.3** |
| Qwen2.5-7B-Base | | | | | | | | | | |
| Base | 62.4 | 5.4 | 2.9 | 26.5 | 16.9 | 28.9 | 11.1 | 70.4 | 42.9 | 29.7 |
| On-policy RL | 76.8 | 14.2 | 9.1 | 46.0 | 34.2 | 41.5 | 29.3 | 86.0 | 48.0 | 42.8 |
| LUFFY | 77.0 | 12.9 | 10.4 | 46.4 | **35.3** | 40.8 | 26.8 | 86.0 | 49.7 | 42.8 |
| QuestA | 78.8 | 14.6 | 13.3 | 47.4 | 33.5 | 41.5 | 30.3 | 86.7 | **51.0** | 44.1 |
| **MENTOR** | **81.4** | **18.3** | **16.5** | **53.1** | 34.9 | **45.2** | **30.8** | **89.6** | 50.2 | **46.7** |

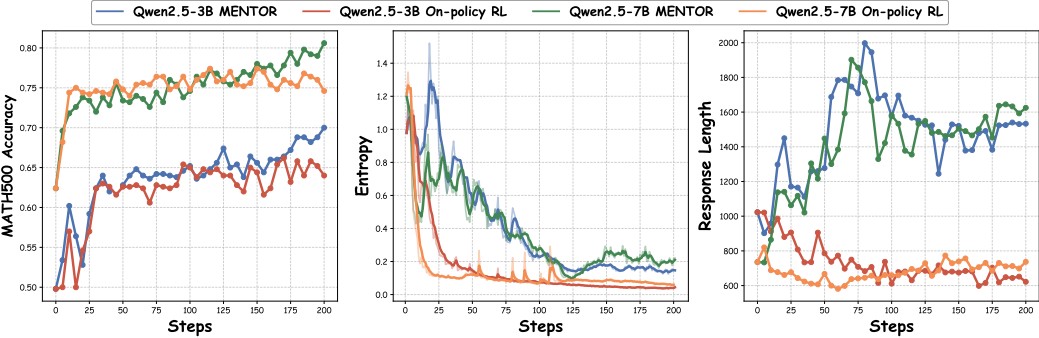

Figure 2: Training dynamics of MENTOR compared with On-policy RL. **MENTOR mitigates entropy collapse, and its response length dynamics reflect a shift from learning to understanding, thereby achieving higher performance.**

## 4.3 TRAINING DYNAMICS

**Entropy dynamics.** Figure 2 compares the training dynamics of On-policy RL and MENTOR in terms of validation accuracy, entropy and response length. Under On-policy RL, entropy collapses rapidly, indicating that the support of the policy exploration space shrinks prematurely to a narrow subset of trajectories. MENTOR enhances exploration diversity through selective expert guidance, thereby slowing down entropy collapse and enabling more persistent exploration throughout training. More importantly, the entropy eventually converges to a slightly higher level than On-policy RL, indicating that the final support set discussed in Section 2 is expanded, which directly translates into stronger final performance.

**Response Length dynamics.** In the early training stage, MENTOR's responses grow in length compared with GRPO. By analyzing rollout samples during training, we find that this rapid growth

---

[2]https://huggingface.co/datasets/open-r1/OpenR1-Math-220k

stems from adopting expert-style reasoning forks such as *verify* and *wait*, the occurrence of which extends the reasoning chain. However, as training progresses, MENTOR's response length gradually declines, consistent with the scheduled reduction of expert advantage. We find that the model starts to distinguish useful tokens (e.g., *verify*) from redundant ones (e.g., *wait*), reflecting a shift from expert-guided to self-driven exploration. Through this selective absorption, the model achieves a more efficient final reasoning pattern, as shown in Appendix G.

## 4.4 THE ANALYSIS OF REASONING PATTERN

To better understand the reasoning patterns induced by different training methods, Figure 3 reports the occurrence rate of high-frequency reasoning tokens, defined as the proportion of trajectories in which the token appears at least once, computed from 500 trajectories on MATH500, which provides a more reliable perspective than individual cases. Detailed case studies are provided in Appendix G.

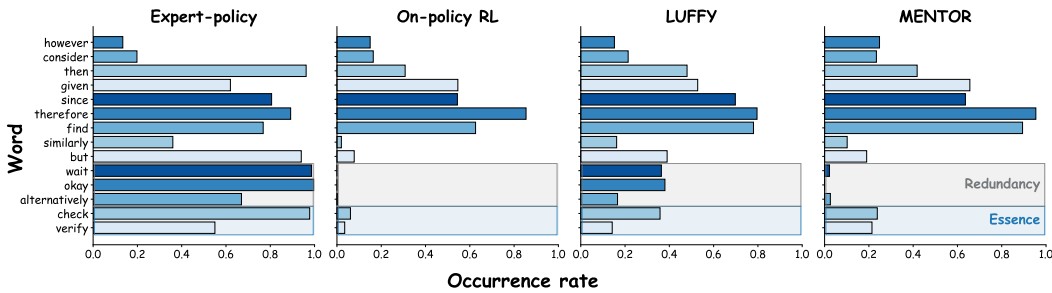

Figure 3: The occurrence rate of high-frequency reasoning tokens under different training methods. **MENTOR absorbs the essence of expert trajectories such as *verify*, while avoiding over-imitation of redundant tokens like *okay* or *wait*.**

**MENTOR achieves selective absorption of expert knowledge.** As shown in Figure 3, although LUFFY successfully incorporate expert knowledge compared with on-policy RL, it tends to imitate indiscriminately. For example, it excessively adopts tokens such as *okay* and *wait*, which leads to overly redundant reasoning. In contrast, MENTOR exhibits a more selective learning process, adopting valuable reasoning tokens such as *verify* and *check* while avoiding preserving redundant ones. This selective learning shows that MENTOR goes beyond surface imitation, effectively absorbing the essence of expert guidance while discarding the redundancy, resulting in an efficient reasoning pattern.

## 4.5 THE ANALYSIS OF REASONING DIVERSITY

To further quantify the impact of different methods on reasoning diversity, we adopt pass@k as the evaluation metric, which is widely used to measure reasoning diversity (Song et al., 2025; Chen et al., 2025). As shown in Figure 4, Pass@32 of On-policy RL stagnates or even declines compared to the Base model, as it can only re-shape behaviors within the original capability, resulting in reduced reasoning diversity. By introducing external expert trajectories, LUFFY and QuestA expand the model's capability boundary and raise pass@k. However, these methods are limited in achieving further improvements in reasoning diversity due to excessive imitation. In contrast, by balancing expert guidance with autonomous exploration, MENTOR achieves a 9.2% average gain in pass@32, indicating a clear enhancement in reasoning diversity.

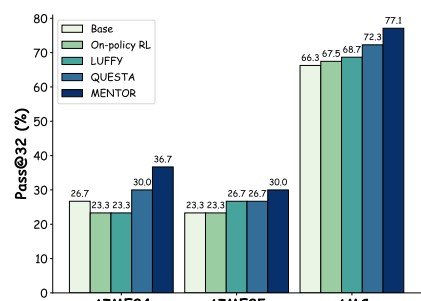

Figure 4: Pass@32 performance of Qwen2.5-7B under different methods. **MENTOR improves the model's reasoning diversity beyond other baselines.**

## 5 RELATED WORK

**Reinforcement Learning for Large Language Models**   Reinforcement learning has recently made significant progress in enhancing the reasoning abilities of LLMs (Jaech et al., 2024; Guo et al., 2025; Team et al., 2025). A central development is Reinforcement Learning from Verifiable Rewards (RLVR), which replaces human feedback signals (Kirk et al., 2024) with automatically checkable objectives such as mathematical verification (Shao et al., 2024) and program execution (Pennino et al., 2025). Such automatically verifiable signals provide reliable supervision and reduce the risk of reward hacking, thereby enabling stable reinforcement learning for complex reasoning tasks (Guo et al., 2025). However, studies also reveal that the gains of RLVR are closely tied to the capability of the base model. For instance, DeepSeek-R1 reports that while RLVR yields remarkable improvements for powerful base models, its benefits become much less pronounced when applied to models with more limited capacity (Guo et al., 2025).

**On-Policy Learning under Expert Guidance**   To improve the effectiveness of RLVR, a line of work incorporates expert trajectories into on-policy RL training. Some approaches directly mix entire expert rollouts with policy rollouts (Yan et al., 2025; Zhang et al., 2025a), while others provide partial prefixes of expert trajectories as hints for continued generation (Liu et al., 2025; Zhang et al., 2025b; Li et al., 2025). These strategies have proven effective in reducing unproductive exploration and stabilizing training. However, imitation of fixed expert trajectories restricts exploration, accelerates entropy collapse (Yan et al., 2025), and ultimately undermines the diversity of reasoning trajectories. In addition, the reduction of diversity is further accelerated by gradient imbalance (Huang et al., 2025), which drives the model to quickly overfit expert trajectories, especially when their reasoning patterns diverge substantially from those of the policy model (Zhang et al., 2025a). Although token-level reweighting has been proposed to alleviate this issue (Yan et al., 2025; Zhang et al., 2025a), the fundamental limitation remains: the exploration is still constrained by the fixed expert trajectories.

**LLM reasoning under guidance**   Generating detailed chains of thought (CoT) has become a central strategy for improving LLM problem-solving performance (Wei et al., 2022). This strategy can be viewed as a form of test-time compute (Muennighoff et al., 2025), where allocating more inference-time FLOPs leads to better performance. Since the quality of the CoT strongly influences final accuracy, a growing body of work focuses on ooptimizing the model's reasoning process. Some approaches leverage the model's own confidence or self-evaluation signals to select higher-value reasoning paths (Yao et al., 2023; Fu et al., 2025; Razghandi et al., 2025). Another line introduces process-reward models that help the model progressively search the output space for more promising CoT trajectories during inference (Snell et al., 2025; Setlur et al., 2024; Zhang et al., 2024; Chen et al., 2024). While these methods improve reasoning by searching within the model's own distribution, their exploration remains inherently bounded by the model's capability. In contrast, our work employs guidance from a more capable expert model, enabling exploration beyond the policy model's native reasoning space and thus providing a stronger mechanism for discovering higher-quality reasoning trajectories.

## 6 CONCLUSION

In this paper, we introduced MENTOR, a powerful framework that enables effective and diverse exploration through selective expert guidance at critical decision points. MENTOR avoids superficial imitation and allows policy model to internalize the essence of expert reasoning strategies. Across challenging benchmarks, our method consistently outperforms strong baselines and significantly improves pass@k performance on complex tasks. These results demonstrate the potential of selective expert guidance to enhance RLVR and suggest promising directions for future research, such as extending the framework to multimodal reasoning or investigating how expert guidance can be provided more effectively.

## 7 ACKNOWLEDGMENTS

This work was supported by Ant Group.

## 8 ETHICS STATEMENT

This work adheres to the ICLR Code of Ethics. In this study, no human subjects or animal experimentation was involved. All datasets used, such as MATH and OpenR1-MATH-220K, were sourced in compliance with relevant usage guidelines, ensuring no violation of privacy. We have taken care to avoid any biases or discriminatory outcomes in our research process. No personally identifiable information was used, and no experiments were conducted that could raise privacy or security concerns. We are committed to maintaining transparency and integrity throughout the research process.

## 9 REPRODUCIBILITY STATEMENT

We have made every effort to ensure that the results presented in this paper are reproducible. All code and datasets have been made publicly available in an anonymous repository to facilitate replication and verification. The experimental setup, including training steps, model configurations, and hardware details, is described in detail in the paper. Furthermore, we will also release the model checkpoints from our main experiments to facilitate future research. The public datasets used in the paper, such as MATH, OpenR1-MATH-220K, are publicly available, ensuring consistent and reproducible evaluation results.

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

## APPENDIX

## A THEORETICAL PROOF

### A.1 PROOF OF EXPLORATION DIVERSITY

**Lemma 2.1** (Policy Distribution under the Expected-Reward Constraint). For a fixed question $q$, let $\mathcal{S}_q = \text{supp}(\pi_\theta(\cdot \mid q))$, $\mathcal{T}^\star = \{\tau \in \mathcal{S}_q : R(\tau) = R_{\max}\}$. Based on the Maximum Entropy

Principle, the policy distribution that attains the largest entropy under the expected-reward constraint $\mathbb{E}_P[R] = C$ takes the Gibbs form

$$P_\lambda(\tau) = \frac{\exp\{\lambda R(\tau)\}}{Z(\lambda)}, \qquad Z(\lambda) = \sum_{\tau' \in \mathcal{S}_q} \exp\{\lambda R(\tau')\}.$$

As the reward constraint $C$ approaches its maximal value $R_{\max}$, the corresponding multiplier $\lambda$ diverges, and all probability mass concentrates on the optimal set $\mathcal{T}^\star$:

$$P_\lambda(\tau) \longrightarrow \begin{cases} \dfrac{1}{|\mathcal{T}^\star|}, & \tau \in \mathcal{T}^\star, \\ 0, & \tau \notin \mathcal{T}^\star. \end{cases}$$

*Proof.* Since the learning objective in Eq.(2) is to maximize expected reward but the exact optimal distribution is unknown, we adopt a Maximum Entropy Principle (Jaynes, 1957). Specifically, we optimize over all probability mass functions $P : \mathcal{S}_q \to [0, 1]$ with $\sum_{\tau \in \mathcal{S}_q} P(\tau) = 1$:

$$\max_P \ H(P) \quad \text{s.t.} \quad \sum_{\tau \in \mathcal{S}_q} P(\tau)R(\tau) = C, \quad \sum_{\tau \in \mathcal{S}_q} P(\tau) = 1, \tag{14}$$

where $H(P) = -\sum_{\tau \in \mathcal{S}_q} P(\tau) \log P(\tau)$ and $C$ is the target expected reward. A standard Lagrangian calculation yields the unique Gibbs-form solution

$$P_\lambda(\tau) = \frac{\exp\{\lambda R(\tau)\}}{Z(\lambda)}, \qquad Z(\lambda) = \sum_{\tau' \in \mathcal{S}_q} \exp\{\lambda R(\tau')\}, \tag{15}$$

for some multiplier $\lambda > 0$ chosen such that $\mathbb{E}_{P_\lambda}[R] = C$.

Define $\phi(\lambda) = \sum_\tau P_\lambda(\tau)R(\tau)$. Then $\phi'(\lambda) = \text{Var}_{P_\lambda}[R] \geq 0$, so $\phi(\lambda)$ is non-decreasing. Moreover, $\lim_{\lambda \to \infty} \phi(\lambda) = R_{\max}$. Hence as $C \uparrow R_{\max}$, we must have $\lambda \to \infty$, and for any $\tau \notin \mathcal{T}^\star$ and $\tau^\star \in \mathcal{T}^\star$,

$$\frac{P_\lambda(\tau)}{P_\lambda(\tau^\star)} = \exp\{-\lambda\left(R_{\max} - R(\tau)\right)\} \longrightarrow 0 \quad (\lambda \to \infty). \tag{16}$$

Thus all probability mass concentrates on $\mathcal{T}^\star$ in the limit.

**Theorem 2.1** (Entropy Upper-Bound Decay with Increasing Expected Reward). In the binary-reward case $R(\tau) \in \{0, 1\}$, let $\mathcal{T}^\star$ be the set of optimal trajectories with $K = |\mathcal{T}^\star|$, $M = |\mathcal{S}_q \setminus \mathcal{T}^\star|$, $N = K + M$. For any expected reward $C \in (0, 1)$, the policy entropy is upper-bounded by

$$H_{\text{ub}}(C) = H_{\text{b}}(C) + C \log K + (1 - C) \log M.$$

where $H_{\text{b}}(C) = -C \log C - (1 - C) \log(1 - C)$.

For $c_2 > c_1$ with $c_1$ larger than the expected reward under the uniform policy on $\text{supp}(\pi_\theta(\cdot \mid q))$ (i.e., $c_1 > \frac{K}{N}$), the entropy upper bound satisfies the single inequality

$$0 \ < \ H_{\text{ub}}(c_1) - H_{\text{ub}}(c_2) = (c_2 - c_1) \log \frac{N}{K} + H_b(c_1) - H_b(c_2),$$

The entropy upper bound necessarily decreases as the expected reward increases, with the amount of inversely proportional to the size $K$ of the optimal trajectory set $\mathcal{T}^\star$.

*Proof.* Let $\mathcal{S}_q$ denote $\text{supp}(\pi_\theta(\cdot \mid q))$. Assume $R(\tau) \in \{0, 1\}$ for all $\tau \in \mathcal{S}_q$ and write

$$\mathcal{T}^\star = \{\tau \in \mathcal{S}_q : R(\tau) = 1\}, \quad K = |\mathcal{T}^\star|, \quad M = |\mathcal{S}_q \setminus \mathcal{T}^\star|, \quad N = K + M.$$

For a fixed target expected reward $C \in (0, 1)$, in the binary case the Gibbs distribution in Eq. (15) is equivalent to

$$\pi_C(\tau) = \begin{cases} \dfrac{C}{K}, & \tau \in \mathcal{T}^\star, \\ \dfrac{1 - C}{M}, & \tau \notin \mathcal{T}^\star. \end{cases} \tag{17}$$

Thus the maximum-entropy solution is uniform over correct trajectories and uniform over incorrect ones, with total mass $C$ and $1 - C$, respectively.

The entropy of $\pi_C$ is

$$H_{\text{ub}}(C) = -\sum_\tau \pi_C(\tau) \log \pi_C(\tau) \tag{18}$$

$$= -C \log \frac{C}{K} - (1 - C) \log \frac{1 - C}{M} \tag{19}$$

$$= H_{\text{b}}(C) + C \log K + (1 - C) \log M, \tag{20}$$

where $H_{\text{b}}(C) = -C \log C - (1 - C) \log(1 - C)$ is the binary entropy. Treating $H(C)$ as a function of $C$, we have

$$H'_{\text{ub}}(C) = -\log C + \log(1 - C) + \log K - \log M = \log \frac{(1 - C)K}{CM}. \tag{21}$$

The critical point satisfies $H'_{\text{ub}}(C) = 0$, which gives

$$\frac{(1 - C)K}{CM} = 1 \quad \Longleftrightarrow \quad C = \frac{K}{K + M} = \frac{K}{N}, \tag{22}$$

i.e., the expected reward under the uniform distribution on $\mathcal{S}_q$. Moreover, $H'_{\text{ub}}(C) < 0$ whenever $C > \frac{K}{N}$, so $H_{\text{ub}}(C)$ is strictly decreasing for $C > \frac{K}{N}$.

Now take $c_1 < c_2$ with $c_1 > \frac{K}{N}$. Since $H_{\text{ub}}$ is strictly decreasing on $(\frac{K}{N}, 1)$, we obtain

$$\Delta H_{\text{ub}}(K) := H_{\text{ub}}(c_1) - H_{\text{ub}}(c_2) > 0.$$

Thus the entropy necessarily drops when the expected reward increases from $c_1$ to $c_2$ in this regime.

Next, for fixed $c_1, c_2$ and $N$, the explicit expression

$$\Delta H_{\text{ub}}(K) = H_{\text{ub}}(c_1) - H_{\text{ub}}(c_2) = [H_{\text{b}}(c_1) - H_{\text{b}}(c_2)] + (c_2 - c_1) \log \frac{N - K}{K} \tag{23}$$

shows that all dependence on $K = |\mathcal{T}^\star|$ is through the factor $\log \frac{N-K}{K}$. Differentiating with respect to $K$ yields

$$\frac{\partial}{\partial K} \Delta H_{\text{ub}}(K) = (c_2 - c_1) \left( -\frac{1}{N - K} - \frac{1}{K} \right) < 0, \tag{24}$$

so $\Delta H_{\text{ub}}(K)$ is strictly decreasing in $K$. Hence, for the same reward increase $c_1 \to c_2$, a larger optimal set $|\mathcal{T}^\star|$ always leads to a smaller entropy drop. In this sense, the entropy loss scales inversely with the size of $\mathcal{T}^\star$, and entropy collapse is slower when the optimal set is larger.

### A.2 Proof of Unbiasedness for Mixed-Policy Rollout

The unbiasedness of speculative sampling is well established in prior work. For completeness, we include a concise proof specialized to our mixed policy $\pi_{\text{mix}}$, confirming that the validation procedure remains unbiased in our setting.

Let the token space be $\mathcal{V}$, and fix a prefix $(q, y_{<t})$ at step $t$. Denote the base policy by

$$p_t(\cdot) = \pi_\theta(\cdot \mid q, y_{<t}), \tag{25}$$

and let $s_t(\cdot) = \pi^*(\cdot \mid q, y_{<t})$ be the expert policy. The mixed policy is obtained by a deterministic ensemble of $(p_t, s_t)$,

$$q_t(\cdot) = \pi_{\text{mix}}(\cdot \mid q, y_{<t}) = \mathcal{M}(p_t(\cdot), s_t(\cdot)), \tag{26}$$

where $\mathcal{M}$ denotes any tokenwise mixing operator that yields a valid distribution on $\mathcal{V}$ (e.g., convex mixing). The validation procedure only depends on $q_t$.

At step $t$, a candidate token $\tilde{y}_t$ is first sampled from $p_t$. It is accepted with probability

$$\alpha_t(\tilde{y}_t) = \min\left(1, \frac{q_t(\tilde{y}_t)}{p_t(\tilde{y}_t)}\right), \tag{27}$$

If rejection occurs, a new token is drawn from the residual distribution on $\mathcal{V}$, defined for the dummy variable $z \in \mathcal{V}$ by

$$r_t(z) = \frac{(q_t(z) - p_t(z))_+}{\sum_{z' \in \mathcal{V}}(q_t(z') - p_t(z'))_+}, \qquad (u)_+ = \max\{u, 0\}. \tag{28}$$

For any possible token $v \in \mathcal{V}$, the probability that it becomes the committed token is therefore

$$\mathbb{P}(y_t = v) = p_t(x) \min\left(1, \frac{q_t(v)}{p_t(v)}\right) + \mathbb{P}(\text{reject})\, r_t(v). \tag{29}$$

The first term equals $\min\{p_t(v), q_t(v)\}$. The rejection probability is

$$\mathbb{P}(\text{reject}) = 1 - \sum_{z \in \mathcal{V}} p_t(z) \min\left(1, \frac{q_t(z)}{p_t(z)}\right) = 1 - \sum_{z \in \mathcal{V}} \min\{p_t(z), q_t(z)\} = \sum_{z \in \mathcal{V}}(q_t(z) - p_t(z))_+, \tag{30}$$

which coincides with the denominator of $r_t(\cdot)$. Consequently, the second term contributes exactly $(q_t(v) - p_t(v))_+$. Combining the two contributions yields

$$\mathbb{P}(y_t = v) = \min\{p_t(v), q_t(v)\} + (q_t(v) - p_t(v))_+ = q_t(v). \tag{31}$$

Thus the distribution of the validated token is exactly the mixed policy $q_t$.

To extend the result to entire speculative sequences, note that at $t = 1$ the marginal distribution is $q_1$. Suppose inductively that the joint distribution of the prefix $y_{<t}$ is $\prod_{j<t} q_j(y_j)$. Conditioning on such a prefix, the above calculation shows that $y_t \sim q_t(\cdot)$. Hence, by induction,

$$\mathbb{P}(y_{1:T} \mid q) = \prod_{t=1}^{T} q_t(y_t) = \prod_{t=1}^{T} \pi_{\text{mix}}(y_t \mid q, y_{<t}), \tag{32}$$

which is identical to direct autoregressive sampling from the mixed policy.

### A.3  PROOF OF AUTOMATIC FILTERING OF MISLEADING EXPERT GUIDANCE

We show that the mixed-policy objective intrinsically filters out misleading or low-quality expert guidance, thereby ensuring robustness even when the expert is weak. For clarity, we rewrite the mixed-policy objective of Eq. (11) in its equivalent expectation form (for analytical convenience, we omit the clipping)

$$\mathcal{J}_{\text{mixed}}(\theta) = \mathbb{E}_{q \sim \mathcal{D},\, \tau \sim \pi_\theta(\cdot|q)}\left[\frac{R(\tau) - \bar{R}}{\text{std}(R)}\right] + \mathbb{E}_{q \sim \mathcal{D},\, \tau \sim \pi_{\text{mix}}(\cdot|q)}\left[\frac{[R(\tau) - \bar{R}]_+}{R_{\text{range}}}\right], \tag{33}$$

where $\bar{R}$ denotes the average reward obtained by on-policy rollouts on the same query $q$, and $[x]_+ = \max(x, 0)$.

The first expectation corresponds to standard GRPO without expert guidance. Thus, we focus on the second term, which represents the contribution of expert guidance. The key observation is that the choice of $[\cdot]_+$ induces an implicit rejection sampling effect. In typical reasoning tasks with binary outcome rewards (correct yields 1, incorrect yields 0), we have

$$[R(\tau) - \bar{R}]_+ = \begin{cases} R(\tau) - \bar{R}, & \text{if } \tau \text{ is correct}, \\ 0, & \text{otherwise}. \end{cases} \tag{34}$$

Consequently, any trajectory, which results in an incorrect answer because of unsuitable or misleading expert guidance, obtains zero advantage and thus contributes no gradient signal, ensuring that such erroneous expert signals are automatically filtered out. We further equivalently rewrite the second term as

$$\mathbb{E}_{q \sim \mathcal{D},\, \tau \sim \pi_{\text{mix}}(\cdot|q)}\left[\frac{[R(\tau) - \bar{R}]_+}{R_{\text{range}}}\right] \tag{35}$$

$$= \int_{\mathcal{T}_{\text{correct}}} \frac{[R(\tau) - \bar{R}]}{R_{\text{range}}} \pi_{\text{mix}}(\tau|q)\, d\tau + \int_{\mathcal{T}_{\text{incorrect}}} 0 \cdot \pi_{\text{mix}}(\tau|q)\, d\tau \tag{36}$$

$$= \mathbb{E}_{q \sim \mathcal{D},\, \tau \sim \pi_{\text{mix}}(\cdot|q),\, \tau \text{ is correct}}\left[\frac{R(\tau) - \bar{R}}{R_{\text{range}}}\right]. \tag{37}$$

where $\mathcal{T}_{\text{correct}}$ and $\mathcal{T}_{\text{incorrect}}$ denote, for a given query $q$, the sets of trajectories that yield correct and incorrect outcomes, respectively.

Eq.(37) shows that the algorithm learns exclusively from effective expert-guided trajectories. Furthermore, the term $(R(\tau) - \bar{R})$ measures the improvement provided by expert guidance over the model's own reasoning, which allows the algorithm to distinguish whether success comes from the model itself or from the expert guidance. Only those expert-guided trajectories that provide genuine improvement beyond the model's baseline ability yield a positive advantage and are consequently reinforced, while guidance that offers no real benefit results in negligible.

In summary, the mixed-policy objective:

- completely suppresses gradient contributions from incorrect expert-guided trajectories, thereby preventing interference from misleading guidance.

- only reinforces expert guidance when it provides measurable improvement over the model's self-generated rollouts.

Even in the extreme case where the expert can provide only misleading guidance, and no correct trajectory can be sampled under such guidance, our method still guarantees a performance lower bound equivalent to standard GRPO, since the second expectation in Eq.(33) becomes zero and thus has no effect on the update.

## B  ALGORITHMIC PROCEDURE OF MENTOR

To complement the main-text description, we provide the full algorithmic procedure of MENTOR in Algorithm 2. The algorithm outlines how mixed-policy expert navigation is integrated into on-policy GRPO training, including the construction of the mixed policy, the dynamic update of the entropy threshold, and the computation of group-wise advantages. For clarity, the pseudocode explicitly separates on-policy rollouts from expert-guided mixed rollouts and highlights how the mixed-policy GRPO objective is optimized at each step.

---

**Algorithm 2** **M**ixed-policy **E**xpert **N**avigation for **T**oken-level **O**ptimization of **R**easoning

---

Given initial policy model $\pi_{\theta_{\text{init}}}$, expert policy model $\pi^*$, task prompts $\mathcal{D}$.
Given hyperparameters $N_1$, $N_2$, $p$, $\mu$, number of total training steps $M$.
Initialize policy model $\pi_\theta \leftarrow \pi_{\theta_{\text{init}}}$.
Initialize entropy threshold $\gamma_p \leftarrow \inf$.
**for** step $= 1 : M$ **do**
    Sample a batch $\mathcal{D}_b$ from $\mathcal{D}$.
    Update old policy model $\pi_{\theta_{\text{old}}} \leftarrow \pi_\theta$.
    Define mixed-policy $\pi_{\text{mix}}$ in Eq. (8) with $\pi_{\theta_{\text{old}}}$, $\pi^*$ and $\gamma_p$
    For each question $q \in \mathcal{D}_b$, sample outputs
        $$\mathcal{G}_{\text{on}} = \{\tau_i\}_{i=1}^{N_1} \sim \pi_{\theta_{\text{old}}}(\cdot \mid q), \quad \mathcal{G}_{\text{mix}} = \{\tau_i\}_{i=1}^{N_2} \sim \pi_{\text{mix}}(\cdot \mid q).$$
    Compute and update the entropy threshold $\gamma_p$ from trajectories in $\mathcal{G}_{\text{on}}$.
    Compute rewards for each trajectory in $\mathcal{G}_{\text{on}} \cup \mathcal{G}_{\text{mix}}$.
    Compute advantages $\hat{A}_{i,t}$ for $\mathcal{G}_{\text{on}}$ and $\mathcal{G}_{\text{mix}}$, using Eq. (12) and Eq. (13), respectively.
    **for** mini step $= 1 : \mu$ **do**
        Update policy parameters $\theta$ by maximizing the Mixed-policy GRPO objective in Eq. (11).
    **end for**
**end for**
**return** $\pi_\theta$

---

## C  EXPERIMENTAL DETAILS

**Platform.**  All of our experiments are conducted on workstations equipped with eight NVIDIA A100 GPUs with 80GB memory, running Ubuntu 22.04.4 LTS and CUDA 12.4.

**System Prompt.** All models trained under MENTOR and other baselines, except QuestA, share the same system prompt for both training and inference:

> **System**
> You are a helpful AI Assistant that provides well-reasoned and detailed responses. You FIRST think about the reasoning process as an internal monologue and then provide the final answer. The reasoning process MUST BE enclosed within <think></think>tags. The final answer MUST BE put in \boxed{}.
> **User**
> {QUESTION}
> **Assistant**

For QuestA, we additionally append "## Hint: Partial Solution" after the QUESTION as a hint section.

**Reward Setting.** For outcome reward, we employ Math-Verify to automatically check whether the final answer inside the "<think>... </think>... \boxed{}" format matches the ground truth, assigning +1 if correct and 0 otherwise. In addition, we introduce a format reward that grants +1 when the response adheres to this format, and 0 if not. The same reward design is applied to MENTOR and all baselines to ensure fairness. For Qwen2.5-7B and Qwen2.5-3B, the weights of outcome reward and format reward are set to 9:1. For LLaMa3.1-8B, however, this ratio is adjusted to 8:2, since the original weighting did not sufficiently enforce format adherence.

**Dataset Details.** For Qwen2.5-7B and Qwen2.5-3B, we use problems from the MATH dataset with difficulty levels 3–5, removing all instances that overlap with the test sets to avoid data leakage. This yields a total of 8,889 training examples. However, for LLaMA3.1-8B, this dataset is too difficult, making the vanilla GRPO algorithm hard to apply. To address this issue, we constructed an easier training set from OpenR1-Math-220K by selecting problems with response lengths shorter than 4K tokens, on which the model could be successfully trained using GRPO. All subsequent methods on LLaMA3.1-8B were trained using this simplified dataset. For each problem, the fixed expert trajectory used in LUFFY and QuestA is generated by DeepSeek-R1.

**Export Model Details.** For Qwen2.5, We adopt OpenR1-Qwen-7B[3] as the expert model in MENTOR, which is trained on a distilled dataset generated by DeepSeek-R1. For LLaMA3.1, the expert model in MENTOR is obtained by further fine-tuning LLaMA3.1-8B-Instruct under the same dataset and setting used for OpenR1-Qwen-7B.

**Training Details.** We conduct all experiments using the EasyR1[4] (Zheng et al., 2025) framework, which employs Verl (Sheng et al., 2024) as the RL training engine and vLLM (Kwon et al., 2023) as the rollout engine. The training setup includes a rollout batch of 128, a learning rate of $1 \times 10^{-6}$, a generation temperature of 1.0, and a higher-clip of 0.28. Each response sequence is up to 8k tokens in length. We perform 8 rollouts per prompt and do not apply KL divergence or entropy regularization (KL Coeff = 0, entropy loss = 0). The mini-batch size is set to 64. For important parameters of MENTOR, $\alpha$ is initialized to 1 and annealed to 0 with a cosine schedule over 120 steps, enabling a smooth transition from expert guidance to autonomous exploration. The number of mixed-policy rollouts is set to 4. For $\gamma_p$, p is chosen as 0.95, corresponding to the 95-th percentile of token-level entropies within each batch. As a special case, $\gamma_p$ is initialized to 999 at the first step.

## D EXPLORING ALTERNATIVE FORMS OF EXPERT GUIDANCE

Beyond the **entropy-based guidance** introduced in the main text, we further investigate several alternative ways of determining where and how expert guidance should be injected during mixed-policy rollout.

---

[3]https://huggingface.co/open-r1/OpenR1-Qwen-7B
[4]https://github.com/hiyouga/EasyR1

**(1) Random guidance.** We begin with a simple baseline that injects expert guidance uniformly at random throughout decoding, without relying on any uncertainty signal or contextual criterion. At each step, the model routes the next-token decision to the expert policy $\pi^*$ with probability $0.2$, and to the base policy $\pi_\theta$ with probability $0.8$. In expectation, this stochastic routing yields the following mixed distribution:

$$\pi_{\text{mix}}(y_t \mid x_{<t}) = 0.8\,\pi_\theta(y_t \mid x_{<t}) + 0.2\,\pi^*(y_t \mid x_{<t}). \tag{38}$$

**(2) Perplexity-based guidance.** Token-level perplexity measures how confused the model is about generating a particular next token. For a token $y_t$ with predicted probability $p_\theta(y_t \mid x_{<t})$, the perplexity is defined as

$$\text{PPL}(t) = \exp\left(-\log p_\theta(y_t \mid x_{<t})\right) = \frac{1}{p_\theta(y_t \mid x_{<t})}. \tag{39}$$

Higher perplexity indicates that the model is more confused about predicting the next token and is more likely to make an error. To leverage this signal, we route the top 20% highest-perplexity tokens to the expert policy. Concretely, let $\tau$ denote the 80th-percentile threshold of token-level perplexity within the sequence, then the mixed policy is defined as:

$$\pi_{\text{mix}}(y_t \mid x_{<t}) = \begin{cases} \pi^*(y_t \mid x_{<t}) & \text{PPL}(t) > \tau, \\ \pi_\theta(y_t \mid x_{<t}) & \text{otherwise.} \end{cases} \tag{40}$$

To provide a direct illustration of how these guidance mechanisms differ in practice, we further analyze the critical tokens generated by expert. Concretely, we use Qwen2.5-7B-Base as the base policy and OpenR1-Qwen-7B as the expert policy, matching the setup used in our main experiments. For each AIME24 query, we decode with temperature (T = 1.0) and apply the three guidance strategies during generation. By aggregating the guidance tokens generated by the expert $\pi^*$ under each strategy, we visualize their distributions in Figure 5.

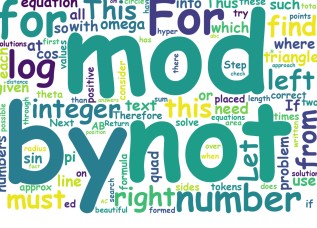 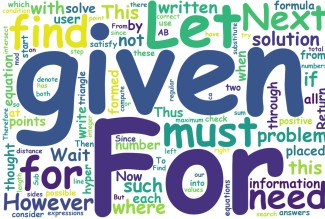

(a) Random guidance      (b) Perplexity-based guidance      (c) Entropy-based guidance

Figure 5: Word-cloud visualizations of expert-generated guidance tokens under different selection strategies.

Compared with random and perplexity-based guidance, entropy-based guidance generates many logical connectors (e.g., wait, however) that, in our experiments, often trigger new reasoning branches and lead to trajectories whose style and structure differ substantially from the model's own reasoning without guidance. By contrast, random and perplexity-based guidance rarely introduce such branching points, and the resulting reasoning trajectories remain close to those produced by the base model alone.

To further validate the downstream impact of different guidance strategies, we follow the main training setup and compare random guidance, perplexity-based guidance, and entropy-based guidance on Qwen2.5-7B-Base.

As shown in Table 2, both random guidance and perplexity-based guidance provide only limited improvement over GRPO, with the latter even occasionally degrading performance. In contrast, entropy-based guidance delivers substantial gains on both MATH and AIME24, indicating that expert guidance is more effective when applied at high-entropy positions.

| Setting | MATH | AIME24 |
|---|---|---|
| GRPO | 76.8 | 14.2 |
| MENTOR (Random guidance) | 77.6 | 14.8 |
| MENTOR (Perplexity-based guidance) | 77.0 | 13.3 |
| MENTOR (Entropy-based guidance) | **81.4** | **18.3** |

Table 2: Impact of different guidance on MENTOR performance.

# E ABLATION STUDY

## E.1 ABLATION OF METHOD COMPONENTS

We analyze the contributions of each component in our methodology, as detailed in Table 3. The observed improvements demonstrate the effectiveness of these components in RL training, with each contributing performance gains on MATH.

| Method | MATH | AIME24 |
|---|---|---|
| Qwen2.5-7B-Base | 62.4 | 5.4 |
| GRPO | 76.8 | 14.2 |
| +Mixed-policy Rollout | 79.4 | 14.6 |
| +Mixed-policy GRPO | **81.4** | **18.3** |

Table 3: Main results of progressive components applied to MENTOR

## E.2 ABLATION OF EXPERT WEIGHT $\alpha$

We also study the effect of the expert weight $\alpha$, comparing the default decaying schedule (from 1 to 0) with several fixed-weight baselines. As shown in Table 5, MENTOR consistently outperforms standard GRPO under all settings, indicating that the framework remains stable and effective under various parameter configurations. However, different values of $\alpha$ induce distinct patterns in how the model acquires and utilizes expert knowledge.

| Setting | MATH | AIME24 |
|---|---|---|
| GRPO (equiv. to $\alpha = 0$) | 76.8 | 14.2 |
| MENTOR (fixed $\alpha = 1.0$) | 78.2 | 13.9 |
| MENTOR (fixed $\alpha = 0.5$) | 80.4 | 16.1 |
| MENTOR (decay $\alpha : 1 \rightarrow 0$) | **81.4** | **18.3** |

Table 4: Effect of expert weights on MENTOR performance.

**Introducing expert knowledge consistently improves model performance across all hyperparameter settings.** Across all hyperparameter configurations, MENTOR consistently surpasses GRPO ($\alpha = 0$), demonstrating that incorporating expert guidance effectively broadens the model's exploration and improves learning stability. This confirms that absorbing expert knowledge is fundamentally beneficial for the training process.

**Beyond injecting expert information, the model must also consolidate and internalize that knowledge.** The experiments reveal that using a lower fixed weight ($\alpha = 0.5$) yields stronger performance than an overly high weight ($\alpha = 1$). This indicates that retaining a degree of autonomy allows the model to selectively reinforce the parts of expert knowledge that are truly useful, rather than relying on it indiscriminately. In other words, preserving autonomy is necessary for genuine understanding rather than rote imitation.

**The decaying schedule achieves the best balance between them.** Early in training, a high mixing weight accelerates learning by leveraging expert guidance; later, as the weight decreases, the model shifts toward autonomous optimization, refining its own strategy and filtering expert signals more effectively. This dynamic adjustment enables the model to both learn from experts and ultimately surpass them, producing the strongest overall performance.

### E.3 ABLATION OF ENTROPY THRESHOLD $\gamma_p$

To assess the sensitivity of MENTOR to the entropy threshold $\gamma_p$, we conduct an ablation study by varying the high-entropy quantile $p$.

| Setting | MATH | AIME24 |
|---|---|---|
| MENTOR ($p = 0.8$) | 80.8 | 17.0 |
| MENTOR ($p = 0.9$) | 80.2 | 17.7 |
| MENTOR ($p = 0.95$) | **81.4** | **18.3** |

Table 5: Effect of entropy threshold $\gamma_p$ on MENTOR performance.

**MENTOR's final performance remains stable across different $\gamma_p$.** As shown in Table 5, the final performance is largely insensitive to the choice of threshold, indicating that MENTOR remains robust across a reasonable range of $\gamma_p$.

## F EFFICIENCY ANALYSIS

To provide a deeper comparison between MENTOR and a range of baselines, including on-policy RL algorithms (GRPO, DAPO) and expert-guided methods (LUFFY, QuestA), we conduct a detailed efficiency analysis during 200 training steps on Qwen2.5-7B-Base, using the same hyperparameters as in the main experiments. For each method, we report the average sequence lengths and the average stage runtimes. Additionally, because different RL methods produce responses of substantially different lengths, we further define an **throughput** metric to ensure fair comparison across methods, which is computed as the average number of tokens that produce gradients per step divided by the average per-step time. The results are shown in Table 6.

| Method | Sequence Length | | Stage Time (s) | | | Total Time (s) | Throughput (tokens/s) |
|---|---|---|---|---|---|---|---|
| | Prompt | Response | Gen | Old | Update | | |
| **On-policy RL** | | | | | | | |
| GRPO | 153 | 828 | 133 | 24 | 87 | 244 | 3474 |
| DAPO | 153 | 833 | 307 | 25 | 92 | 424 | 2011 |
| **Expert-guided RL** | | | | | | | |
| LUFFY | 153 | 2902 | 270 | 60 | 230 | 560 | 5306 |
| QuestA | 510 | 711 | 142 | 31 | 117 | 290 | 2510 |
| MENTOR | 153 | 1751 | 404 | 48 | 175 | 627 | 2860 |

Table 6: Efficiency analysis of different methods. Here, **Gen**, **Old** and **Update** denote respectively the generation (rollout) phase, the computing of the logits of $\pi_{\text{old}}$, and the model update phase in the Verl framework.

**MENTOR achieves the highest performance with only moderate and acceptable additional training overhead.** Since different methods generate responses of different lengths, we mainly rely on throughput for a fair comparison. Compared with on-policy RL methods, MENTOR reaches 2860 tokens/s, between GRPO (3474) and DAPO (2011), because DAPO often performs two or three full generation phases to collect enough samples, while MENTOR's mixed-policy rollouts are more efficient than repeated full generations. For expert-guided methods, LUFFY shows high throughput

partly because it mixes in a full offline expert trajectory of about 6k tokens, which increases the number of processed tokens. From the perspective of the algorithmic design, the throughput of LUFFY's newly generated rollout data should be close to that of GRPO (3474). QuestA concatenates expert segments into the input, creating longer prompts that slightly reduce training throughput. Compared with these approaches, MENTOR achieves the highest final performance, and although it relies on expert guidance during the rollout stage, the additional overhead remains acceptable.

## G  CASE STUDY

To complement the aggregate analysis in Figure 3, we provide representative trajectory-level cases in this section. These examples illustrate how different training methods influence the emergence of reasoning tokens such as *verify*, *check*, and *wait*. By examining full reasoning traces, we highlight qualitative differences between methods, offering additional context to the quantitative analysis in the main text.

We find that the incorporation of expert policy indeed shapes the final reasoning patterns of the models, with different methods leading to distinct outcomes.

- **From a stylistic perspective,** QuestA and MENTOR exhibit reasoning traces more aligned with on-policy RL, whereas LUFFY shows a closer resemblance to the expert's style. This difference arises because LUFFY emphasizes direct imitation of expert demonstrations, often reproducing surface-level tokens such as *wait* and *okay*, while QuestA and MENTOR encourage the model to explore autonomously while using expert guidance, thereby preserving a degree of its own stylistic characteristics.

- **From a content perspective,** MENTOR and LUFFY successfully acquire specific reasoning modes from the expert, such as the use of *verify*, which are absent in the base model. In contrast, QuestA, despite leveraging expert policy, does not exhibit the adoption of *verify*. Our analysis suggests this limitation stems from the design choice of using only the first half of the expert trajectory as hints, while tokens like *verify* typically appear toward the end of trajectories, thereby restricting their adoption.

---

**An example of On-policy RL**

**QUESTION**

Let $x_1$, $x_2$, $x_3$, $y_1$, $y_2$, and $y_3$ be real numbers such that

$$(x_1 - x_2)^2 + (y_1 - y_2)^2 = 9,$$
$$(x_1 - x_3)^2 + (y_1 - y_3)^2 = 16,$$
$$(x_2 - x_3)^2 + (y_2 - y_3)^2 = 25.$$

Find

$$\begin{vmatrix} x_1 & y_1 & 1 \\ x_2 & y_2 & 1 \\ x_3 & y_3 & 1 \end{vmatrix}^2.$$

**RESPONSE**

```
<think>
```
To solve for $\begin{vmatrix} x_1 & y_1 & 1 \\ x_2 & y_2 & 1 \\ x_3 & y_3 & 1 \end{vmatrix}^2$ , we start by defining the determinant of the matrix:

$$D = \begin{vmatrix} x_1 & y_1 & 1 \\ x_2 & y_2 & 1 \\ x_3 & y_3 & 1 \end{vmatrix}.$$

The determinant $D$ can be expanded as follows:

$$D = x_1 \begin{vmatrix} y_2 & 1 \\ y_3 & 1 \end{vmatrix} - y_1 \begin{vmatrix} x_2 & 1 \\ x_3 & 1 \end{vmatrix} + \begin{vmatrix} x_2 & y_2 \\ x_3 & y_3 \end{vmatrix}.$$

---

This expands to:
$$D = x_1(y_2 - y_3) - y_1(x_2 - x_3) + (x_2y_3 - x_3y_2).$$

So, we have:
$$D = x_1y_2 - x_1y_3 - y_1x_2 + y_1x_3 + x_2y_3 - x_3y_2.$$

We need to find $D^2$. To do this, we will use the given distances between the points $(x_1, y_1)$, $(x_2, y_2)$, and $(x_3, y_3)$. The distances are:
$$(x_1 - x_2)^2 + (y_1 - y_2)^2 = 9,$$
$$(x_1 - x_3)^2 + (y_1 - y_3)^2 = 16,$$
$$(x_2 - x_3)^2 + (y_2 - y_3)^2 = 25.$$

These distances suggest that the points $(x_1, y_1)$, $(x_2, y_2)$, and $(x_3, y_3)$ form a right triangle with sides of lengths 3, 4, and 5. Without loss of generality, we can place the points in a coordinate system where the distances are satisfied. One possible configuration is:
$$(x_1, y_1) = (0, 0), \quad (x_2, y_2) = (3, 0), \quad (x_3, y_3) = (0, 4).$$

Now, we calculate the determinant $D$ for these points:
$$D = \begin{vmatrix} 0 & 0 & 1 \\ 3 & 0 & 1 \\ 0 & 4 & 1 \end{vmatrix}.$$

Expanding this determinant, we get:
$$D = 0 \cdot \begin{vmatrix} 0 & 1 \\ 4 & 1 \end{vmatrix} - 0 \cdot \begin{vmatrix} 3 & 1 \\ 0 & 1 \end{vmatrix} + 1 \cdot \begin{vmatrix} 3 & 0 \\ 0 & 4 \end{vmatrix} = 1 \cdot (3 \cdot 4 - 0 \cdot 0) = 12.$$

So, $D = 12$, and we need to find $D^2$:
$$D^2 = 12^2 = 144.$$

Thus, the answer is:
$$\boxed{144}.$$

```
</think>
```
$\boxed{144}$

---

**An example of MENTOR**

**QUESTION**
Let $x_1$, $x_2$, $x_3$, $y_1$, $y_2$, and $y_3$ be real numbers such that
$$(x_1 - x_2)^2 + (y_1 - y_2)^2 = 9,$$
$$(x_1 - x_3)^2 + (y_1 - y_3)^2 = 16,$$
$$(x_2 - x_3)^2 + (y_2 - y_3)^2 = 25.$$

Find
$$\begin{vmatrix} x_1 & y_1 & 1 \\ x_2 & y_2 & 1 \\ x_3 & y_3 & 1 \end{vmatrix}^2.$$

**RESPONSE**
```
<think>
```
To find $\begin{vmatrix} x_1 & y_1 & 1 \\ x_2 & y_2 & 1 \\ x_3 & y_3 & 1 \end{vmatrix}^2$, we start by recalling that the determinant of the given matrix is the area of the triangle formed by the points $(x_1, y_1)$, $(x_2, y_2)$, and $(x_3, y_3)$ multiplied by 2.

Let's denote the determinant by $D$, so $D = \begin{vmatrix} x_1 & y_1 & 1 \\ x_2 & y_2 & 1 \\ x_3 & y_3 & 1 \end{vmatrix}$. Then, $D^2$ is the square of the area of the triangle.

The area $A$ of the triangle can be found using the formula for the area of a triangle with vertices $(x_1, y_1)$, $(x_2, y_2)$, and $(x_3, y_3)$:

$$A = \tfrac{1}{2} \left| x_1(y_2 - y_3) + x_2(y_3 - y_1) + x_3(y_1 - y_2) \right|$$

Thus, $D^2 = 4A^2$. To find $A^2$, we need to use the given distances between the points. The distances are given by the equations:

$$(x_1 - x_2)^2 + (y_1 - y_2)^2 = 9,$$
$$(x_1 - x_3)^2 + (y_1 - y_3)^2 = 16,$$
$$(x_2 - x_3)^2 + (y_2 - y_3)^2 = 25.$$

These equations represent the squared lengths of the sides of the triangle. Let's denote the points as $A = (x_1, y_1)$, $B = (x_2, y_2)$, and $C = (x_3, y_3)$. The distances $AB = 3$, $AC = 4$, and $BC = 5$. Notice that $3^2 + 4^2 = 5^2$, which suggests that triangle $ABC$ is a right triangle with $\angle A = 90°$.

**To verify**, let's use the coordinates to check if the points form a right triangle. Assume $A = (x_1, y_1)$, $B = (x_2, y_2)$, and $C = (x_3, y_3)$. The squared distances are:

$$AB^2 = (x_1 - x_2)^2 + (y_1 - y_2)^2 = 9,$$
$$AC^2 = (x_1 - x_3)^2 + (y_1 - y_3)^2 = 16,$$
$$BC^2 = (x_2 - x_3)^2 + (y_2 - y_3)^2 = 25.$$

Since $AB = 3$, $AC = 4$, and $BC = 5$, and $3^2 + 4^2 = 5^2$, the triangle is indeed a right triangle with $\angle A = 90°$.

The area $A$ of a right triangle is given by half the product of the legs:

$$A = \tfrac{1}{2} \times AB \times AC = \tfrac{1}{2} \times 3 \times 4 = 6.$$

Therefore, $D^2 = 4A^2 = 4 \times 6^2 = 144$.

So, the answer is $\boxed{144}$.

</think>

To find $\begin{vmatrix} x_1 & y_1 & 1 \\ x_2 & y_2 & 1 \\ x_3 & y_3 & 1 \end{vmatrix}^2$, we start by recalling that the determinant of the matrix is the area of the triangle formed by the points $(x_1, y_1)$, $(x_2, y_2)$, and $(x_3, y_3)$ multiplied by 2. Let's denote the determinant by $D$, so $D = \begin{vmatrix} x_1 & y_1 & 1 \\ x_2 & y_2 & 1 \\ x_3 & y_3 & 1 \end{vmatrix}$. Then, $D^2$ is the square of the area of the triangle.

Given the distances between the points:

$$(x_1 - x_2)^2 + (y_1 - y_2)^2 = 9,$$
$$(x_1 - x_3)^2 + (y_1 - y_3)^2 = 16,$$
$$(x_2 - x_3)^2 + (y_2 - y_3)^2 = 25,$$

we recognize these as the squared lengths of the sides of a triangle with sides $AB = 3$, $AC = 4$, and $BC = 5$. Since $3^2 + 4^2 = 5^2$, the triangle is a right triangle with $\angle A = 90°$. The area $A$ of a right triangle is given by:

$$A = \tfrac{1}{2} \times AB \times AC = \tfrac{1}{2} \times 3 \times 4 = 6.$$

The determinant $D$ is twice the area of the triangle:

$$D = 2A = 12.$$

Therefore,

$$D^2 = 12^2 = 144.$$

Thus, the final answer is:

$$\boxed{144}.$$

---

### An example of LUFFY

**QUESTION**

Let $x_1$, $x_2$, $x_3$, $y_1$, $y_2$, and $y_3$ be real numbers such that

$$(x_1 - x_2)^2 + (y_1 - y_2)^2 = 9,$$
$$(x_1 - x_3)^2 + (y_1 - y_3)^2 = 16,$$
$$(x_2 - x_3)^2 + (y_2 - y_3)^2 = 25.$$

Find

$$\begin{vmatrix} x_1 & y_1 & 1 \\ x_2 & y_2 & 1 \\ x_3 & y_3 & 1 \end{vmatrix}^2.$$

**RESPONSE**

`<think>`

Okay, so I've got this problem here about finding the square of the determinant of a matrix given some conditions on the coordinates. Let me try to break it down step by step.

First, the problem gives me six real numbers: $x_1, x_2, x_3, y_1, y_2, y_3$. It also provides three equations involving these numbers:

1. $(x_1 - x_2)^2 + (y_1 - y_2)^2 = 9$ 2. $(x_1 - x_3)^2 + (y_1 - y_3)^2 = 16$ 3. $(x_2 - x_3)^2 + (y_2 - y_3)^2 = 25$

I need to find the square of the determinant of the matrix:

$$\begin{vmatrix} x_1 & y_1 & 1 \\ x_2 & y_2 & 1 \\ x_3 & y_3 & 1 \end{vmatrix}$$

Hmm, okay. Let me recall that the determinant of a 3x3 matrix can be calculated using the formula:

$$\det = x_1(y_2 \cdot 1 - y_3 \cdot 1) - y_1(x_2 \cdot 1 - x_3 \cdot 1) + 1(x_2 y_3 - x_3 y_2)$$

Simplifying that, it becomes:

$$\det = x_1(y_2 - y_3) - y_1(x_2 - x_3) + x_2 y_3 - x_3 y_2$$

So, the square of the determinant is:

$$\det^2 = [x_1(y_2 - y_3) - y_1(x_2 - x_3) + x_2 y_3 - x_3 y_2]^2$$

But wait, maybe there's a better way to approach this. I remember that the determinant of a matrix formed by coordinates of points in a plane is related to the area of the triangle formed by those points. Specifically, the absolute value of the determinant is twice the area of the triangle. So, perhaps I can use that.

Let me denote the points as $A(x_1, y_1)$, $B(x_2, y_2)$, and $C(x_3, y_3)$. The determinant is twice the area of triangle ABC. So, if I can find the area, I can square it.

The area of triangle ABC can be found using the formula:

$$\text{Area} = \frac{1}{2} |x_1(y_2 - y_3) + x_2(y_3 - y_1) + x_3(y_1 - y_2)|$$

But wait, that's the same as the determinant divided by 2. So, the determinant is twice the area. Therefore, the square of the determinant is four times the square of the area.

So, I need to find the area of triangle ABC. Let me denote the area as $A$. Then, the square of the determinant is $4A^2$.

Now, the area $A$ can be found using the distance formula. The three given equations are the squared distances between the points. Let me denote the points as $A$, $B$, and $C$. So, $AB^2 = 9$, $AC^2 = 16$, and $BC^2 = 25$. Therefore, $AB = 3$, $AC = 4$, and $BC = 5$.

Hmm, interesting. So, the sides of the triangle are 3, 4, and 5. That's a right triangle because $3^2 + 4^2 = 5^2$. So, the area of a right triangle is $\frac{1}{2} \times$ base $\times$ height. In this case, the base and height are 3 and 4, so the area is $\frac{1}{2} \times 3 \times 4 = 6$.

Therefore, the square of the area is $6^2 = 36$. So, the square of the determinant is $4 \times 36 = 144$.

Wait, let me **double-check**. The area is 6, so the determinant squared is $4 \times 6^2 = 4 \times 36 = 144$. That seems correct.

But let me make sure I didn't make a mistake in the area calculation. The sides are 3, 4, 5, which form a right triangle. The area is indeed $\frac{1}{2} \times 3 \times 4 = 6$. So, the square is 36, and the determinant squared is 144.

I think that's it. So, the final answer is 144.

`</think>`

The square of the determinant is $\boxed{144}$.

---

## An example of QuestA

**QUESTION**

Let $x_1$, $x_2$, $x_3$, $y_1$, $y_2$, and $y_3$ be real numbers such that

$$(x_1 - x_2)^2 + (y_1 - y_2)^2 = 9,$$
$$(x_1 - x_3)^2 + (y_1 - y_3)^2 = 16,$$
$$(x_2 - x_3)^2 + (y_2 - y_3)^2 = 25.$$

Find

$$\begin{vmatrix} x_1 & y_1 & 1 \\ x_2 & y_2 & 1 \\ x_3 & y_3 & 1 \end{vmatrix}^2.$$

**RESPONSE**

`<think>`

Given the equations:

$$(x_1 - x_2)^2 + (y_1 - y_2)^2 = 9,$$
$$(x_1 - x_3)^2 + (y_1 - y_3)^2 = 16,$$
$$(x_2 - x_3)^2 + (y_2 - y_3)^2 = 25,$$

we need to find $\begin{vmatrix} x_1 & y_1 & 1 \\ x_2 & y_2 & 1 \\ x_3 & y_3 & 1 \end{vmatrix}^2.$

Let's denote the determinant by $D$:

$$D = \begin{vmatrix} x_1 & y_1 & 1 \\ x_2 & y_2 & 1 \\ x_3 & y_3 & 1 \end{vmatrix}.$$

The square of the determinant $D^2$ can be found using the formula for the square of a determinant of a $3 \times 3$ matrix. However, we can use a geometric interpretation to simplify the problem.

Consider the points $A = (x_1, y_1)$, $B = (x_2, y_2)$, and $C = (x_3, y_3)$ in the coordinate plane. The given equations represent the squared distances between these points:

$$AB^2 = 9,$$
$$AC^2 = 16,$$
$$BC^2 = 25.$$

These distances suggest that the points $A$, $B$, and $C$ form a right triangle with $BC$ as the hypotenuse. This is because $9 + 16 = 25$, which satisfies the Pythagorean theorem.

Without loss of generality, let's assume $B$ is the right angle, so $A$, $B$, and $C$ form a right triangle with $BC$ as the hypotenuse. This assumption simplifies our calculations.

The area of the triangle $ABC$ can be calculated using the distances given. The area $K$ of a triangle with vertices $(x_1, y_1)$, $(x_2, y_2)$, and $(x_3, y_3)$ can also be found using the determinant formula for the area of a triangle:

$$K = \frac{1}{2} |x_1(y_2 - y_3) + x_2(y_3 - y_1) + x_3(y_1 - y_2)|.$$

> The square of the determinant $D$ is related to the area of the triangle. Specifically, the square of the determinant of the matrix formed by the coordinates of the points is four times the square of the area of the triangle:
> $$D^2 = 4K^2.$$
>
> To find $K^2$, we use the fact that the points form a right triangle. The legs of the triangle are the distances $AB$ and $AC$, and the hypotenuse is $BC$. The area $K$ of the right triangle can be calculated as:
> $$K = \frac{1}{2} \cdot AB \cdot AC = \frac{1}{2} \cdot \sqrt{9} \cdot \sqrt{16} = \frac{1}{2} \cdot 3 \cdot 4 = \frac{1}{2} \cdot 12 = 6.$$
>
> Thus, the square of the area $K^2$ is:
> $$K^2 = 6^2 = 36.$$
>
> Therefore, the square of the determinant $D^2$ is:
> $$D^2 = 4K^2 = 4 \cdot 36 = 144.$$
>
> `</think>`
> The final answer is:
> $$\boxed{144}.$$

## H LLM USAGE

Large Language Models (LLMs) were used to aid in the writing and polishing of the manuscript. Specifically, we used an LLM to assist in refining the language, improving readability, and ensuring clarity in various sections of the paper. The model helped with tasks such as sentence rephrasing, grammar checking, and enhancing the overall flow of the text.

It is important to note that the LLM was not involved in the ideation, research methodology, or experimental design. All research concepts, ideas, and analyses were developed and conducted by the authors. The contributions of the LLM were solely focused on improving the linguistic quality of the paper, with no involvement in the scientific content or data analysis.

The authors take full responsibility for the content of the manuscript, including any text generated or polished by the LLM. We have ensured that the LLM-generated text adheres to ethical guidelines and does not contribute to plagiarism or scientific misconduct.

