# OpenReview forum: "Selective Expert Guidance for Effective and Diverse Exploration in Reinforcement Learning of LLMs"
_ICLR.cc/2026/Conference — ICLR 2026 Poster_

### Official Review · Reviewer_QY8s · 2025-10-30

**Soundness:** 2
**Presentation:** 3
**Contribution:** 2
**Rating:** 4
**Confidence:** 3

**Summary:**

This paper proposes MENTOR, a reinforcement learning framework for large language models that enhances reasoning by providing expert guidance only at critical decision points rather than on entire reasoning paths. This approach promotes both effective and diverse exploration, enabling models to capture strategic essence rather than superficial imitation, which leads to superior performance compared to existing RLVR methods.

**Strengths:**

- This paper is easy to read and easy to follow
- This paper focuses on an important topic in large language model reasoning, aiming at enhancing the exploration capabilities of existing LLMs via some specially designed approaches
- The authors provide codes in the main text, which enhances the reliability of the results reported in this paper
- The experiments are carried out using both LLaMa models and Qwen models with different model sizes, which shows the generality of the proposed method to some extent

**Weaknesses:**

- (major) I do not see the necessity of Definitions 2.1 and 2.2. If the authors would like to claim that the optimal trajectory can be rarely covered, and hence raises two critical issues in the RLVR training, there is no need to state Definition 2.1 and 2.2 since the readers can generally get that. The organization of this paper can be improved
- (major) No ablation study and parameter study can be found either in the main text or the appendix. This makes it hard to figure out which component (mixed-policy rollout and mixed-policy GRPO) contributes most to the performance of MENTOR. It is also hard to check how sensitive MENTOR is to the introduced hyperparameters, e.g., the initial value of $\alpha$ in Eq. 11, how different decay schedules of $\alpha$ affect the performance, how to set $p$ in $\gamma_p$, etc.
- (major) Despite the fact that the authors propose a method to accelerate mixed-policy rollout, the entire process of MENTOR is still both time-consuming and memory-consuming compared to other methods like QuestA. The authors ought to show the training efficiency of MENTOR against baseline methods
- (major) It is somewhat strange to me that the authors use different expert models for different base LLMs, e.g., OpenR1-Qwen-7B model for Qwen models and finetuned LLaMA3.1-8B-Instruct model for LLaMa models. Is there any reason that you do not use a fixed expert model? Why do you use the Instruct model for LLaMa here? It is unclear whether the proposed method would fail if one adopts other LLMs as the base model, e.g., deepseek-R1 as the expert model.
- (major) It is also unclear whether the proposed method still works if the expert model is only marginally stronger than the base model, or if the expert model is too strong (e.g., Qwen3-max), and whether the proposed method can still work. Note that QuestA conducts experiments using some strong base LLMs. The authors commented that the advantages of QuestA rely heavily on the vanilla performance of the base models, while it raises questions on whether MENTOR can work if it is adopted upon some strong base models (e.g.,  Nemotron-1.5B)
- (minor) The related work part can be significantly improved. LLM reasoning and LLM reasoning under guidance are fast-growing research areas, and there are numerous papers that do similar things. The authors should include more discussion in the manuscript
- (minor) No full pseudo-code for MENTOR can be found. Algorithm 1 just states part of the MENTOR algorithm
- (minor) There are numerous minor issues in the manuscript. I just list some of them below, please double-check your manuscript,
  - Line 019, *we argue that the expert only needs to provide guidance only at critical*, two *only* in this sentence
  - Line 322, two QuestA

**Questions:**

- Why do you still adopt trajectories from the current model in Eq. 9? Any insights here?
- In Eq. 11, you write $A_i = \dfrac{[R\_i - mean(R\_j)]\_+}{R\_{\rm range}}$ rather than $\dfrac{[R\_i - mean(R\_j)]\_+}{{\rm std}(R\_j)}$, why? Why only normalize the advantages from the mixed policy into [0, 1] but use Eq. 10 directly from GRPO?
- Can the proposed MENTOR still work when the base LLM is strong?

---

> ### Author Response · Authors · 2025-11-23
> **Response to Reviewer QY8s (1/6)**
>
> Thank you for your valuable feedback to help us improve our paper. We have revised our paper based on your feedback. We detail our response below and please kindly let us know if our response addresses your concerns.
>
> ---
>
> >**Q1:** (major) I do not see the necessity of Definitions 2.1 and 2.2. If the authors would like to claim that the optimal trajectory can be rarely covered, and hence raises two critical issues in the RLVR training, there is no need to state Definition 2.1 and 2.2 since the readers can generally get that. The organization of this paper can be improved
>
>
> We thank the reviewer for the suggestion. While the effectiveness issue of exploration is indeed intuitive, Definitions 2.1 and 2.2 serve a purpose beyond stating that optimal trajectories are rarely covered. They allow us to formally characterize the relationship between the optimal-trajectory set $\mathcal{T}^\star$ and the exploration diversity during RLVR training, which would be difficult to express precisely without these definitions.
>
>
> To strengthen this connection, we extend the analysis in Appendix A.1.1 and introduce a new theoretical result, **Theorem 2.1**, which shows that as the expected reward increases, the policy’s entropy upper bound necessarily decreases at a rate governed by the size of $\mathcal{T}^\star$.
>
> Finally, we have revised the structure of Sec. 2.2 to make the role of Definitions 2.1 and 2.2 clearer and reduce potential confusion for readers.

---

> ### Author Response · Authors · 2025-11-23
> **Response to Reviewer QY8s (2/6)**
>
> >**Q2:** (major) No ablation study and parameter study can be found either in the main text or the appendix. This makes it hard to figure out which component (mixed-policy rollout and mixed-policy GRPO) contributes most to the performance of MENTOR. It is also hard to check how sensitive MENTOR is to the introduced hyperparameters, e.g., the initial value of $\alpha$ in Eq. 11, how different decay schedules of $\alpha$ affect the performance, how to set  $p$ in $\gamma_p$, etc.
>
> We thank the reviewer for pointing this out. To address these concerns, we have conducted a comprehensive set of ablation studies covering
> * (i) MENTOR’s core components,
> * (ii) key hyperparameters—including the expert weight $\alpha$ and the entropy threshold $\gamma_p$
> * (iii) alternative mixing strategies.
>
> All results are now included in Appendix A.3–A.4 of the revised paper.
>
> ### **Ablation on components of MENTOR**
> **Table R1:** Main results of **component ablation** of MENTOR
> | Method                 | MATH | AIME24 |
> |------------------------|:----:|:------:|
> | Qwen2.5-7B-Base        | 62.4 |  5.4   |
> | GRPO                   | 76.8 | 14.2   |
> | +Mixed-policy Rollout  | 79.4 | 14.6   |
> | +Mixed-policy GRPO     | **81.4** | **18.3** |
>
> **Each component contributes.** The observed improvements demonstrate the effectiveness of these components in RL training, with each contributing performance gains on MATH.
>
> ### **Ablation on the Expert weight** $\alpha$
> **Table R2:** Ablation on the **Expert weight $\alpha$** in MENTOR
> | Setting                       | MATH | AIME24 |
> |------------------------------|:----:|:------:|
> | GRPO (equiv. to $\alpha$ = 0)       | 76.8 | 14.2   |
> | MENTOR (fixed $\alpha$ = 1.0)       | 78.2 | 13.9   |
> | MENTOR (fixed $\alpha$ = 0.5)       | 80.4 | 16.1   |
> | MENTOR (decay $\alpha$ : 1 → 0)     | **81.4** | **18.3** |
>
> We identified the following findings in Ablation on the Expert weight $\alpha$:
> 1. **Introducing expert knowledge consistently improves performance.**
> Across all settings, MENTOR outperforms GRPO ($\alpha = 0$), showing that adding expert information reliably boosts results.
>
> 2. **Moderate expert weighting outperforms an excessively high weight.**
> Although both $\alpha = 0.5$ and $\alpha = 1$ use expert guidance, the former achieves higher scores, indicating that moderate mixing, which preserves some degree of autonomous exploration, leads to better performance than excessively relying on expert guidances.
>
> 3. **A decaying schedule delivers the best results.**
> The decaying schedule achieves the highest performance, suggesting that gradually reducing the expert weight enables a smooth shift from guided learning to autonomous optimization.
>
> ### **Ablation on the Entropy Threshold** $\gamma_p$
> **Table R3:** Ablation on the **Entropy Threshold** $\gamma_p$ in MENTOR
> | Setting            |  MATH | AIME24 |
> |--------------------|:-----:|:------:|
> | MENTOR *(p = 0.8)* | 80.8  | 17.0   |
> | MENTOR *(p = 0.9)* | 80.2  | 17.7   |
> | MENTOR *(p = 0.95)*| **81.4** | **18.3** |
>
> **MENTOR’s final performance remains stable across different $\gamma_p$.** Table R3 shows that the final results vary only slightly with the threshold, demonstrating that MENTOR is robust to the choice of $\gamma_p$.

---

> ### Author Response · Authors · 2025-11-23
> **Response to Reviewer QY8s (3/6)**
>
> ### **Exploring Alternative Forms of Mixed-policy**
> We adopt entropy as the selection principle because prior studies[1][2] show that high-entropy tokens typically are critical decision points in the reasoning process.
>
> We also agree with the reviewers that examining alternative selection strategies is important for assessing whether entropy is indeed the most effective choice compared with other options. We further investigate several alternative ways, including random/uniform guidance (both are equivalent) and based-perplexity guidance. We have revised the paper and put these details in Appendix A.3.
>
> 1. **Random/uniform guidance.**
> This baseline injects expert tokens uniformly at random with a fixed probability, yielding the mixed policy
> $$
> \pi_{\text{mix}}(y_t \mid x_{<t}) = 0.8\pi_\theta(y_t \mid x_{<t}) + 0.2\pi^{*}(y_t \mid x_{<t}).
> $$
>
> 2. **Perplexity-based guidance.**
> This strategy routes tokens with perplexity above the 80th-percentile threshold $\tau$ to the expert, producing the policy
> $$
> \pi_{\text{mix}}(y_t \mid x_{<t})=
> \begin{cases}
> \pi^{*}(y_t \mid x_{<t}) & \mathrm{PPL}(t) > \tau,\\
> \pi_{\theta}(y_t \mid x_{<t}) & \text{otherwise}.
> \end{cases}
> $$
>
> To provide a direct illustration of how these guidance mechanisms differ in practice, we further visualize the critical tokens generated by the expert in Appendix A.3.
>
> We identified the following findings:
>
> 1. **Entropy-based guidance frequently introduces logical connectors** (e.g., *wait*, *however*), which often trigger new reasoning branches and produce trajectories whose style and structure deviate significantly from the model’s unguided reasoning.
> 2. **Random and perplexity-based guidance seldom create such branching points**, yielding trajectories that stay close to those generated by the base model.
> 3. **Entropy-based guidance yields the largest performance gains**, substantially outperforming random and perplexity-based guidance on both MATH and AIME24 as shown in Table R4, indicating that expert guidance is most effective when applied at high-entropy positions.
>
>
> **Table R4:** Impact of **different guidance** on MENTOR performance.
> | Setting                               | MATH | AIME24 |
> |----------------------------------------|:----:|:------:|
> | GRPO                                  | 76.8 | 14.2   |
> | MENTOR (Random guidance)              | 77.6 | 14.8   |
> | MENTOR (Perplexity-based guidance)    | 77.0 | 13.3   |
> | MENTOR (Entropy-based guidance)       | **81.4** | **18.3** |
>
>
> [1] Beyond the 80/20 Rule: High-Entropy Minority Tokens Drive Effective Reinforcement Learning for LLM Reasoning \
> [2] Blending Supervised and Reinforcement Fine-Tuning with Prefix Sampling

---

> ### Author Response · Authors · 2025-11-23
> **Response to Reviewer QY8s (4/6)**
>
> >**Q3:** (major) Despite the fact that the authors propose a method to accelerate mixed-policy rollout, the entire process of MENTOR is still both time-consuming and memory-consuming compared to other methods like QuestA. The authors ought to show the training efficiency of MENTOR against baseline methods
>
> We appreciate the reviewer’s concern regarding the training efficiency. To address this point, we conduct a dedicated efficiency analysis during 200 training steps on Qwen2.5-7B-Base (see Appendix.A.5 for details). Additionally, because different RL methods produce responses of substantially different lengths, we further define an **throughput** metric to ensure fair comparison across methods, which is computed as the average number of tokens that produce gradients per step divided by the average per-step time. All results are summarized in Table R5
>
> **Table R5:** Efficiency analysis of different methods. Here, Gen, Old and Update denote respectively the generation (rollout) phase, the computing of the logits of $\pi_\text{old}$, and the model update phase in the Verl framework.
> | Method | Prompt_len | Resp_len | Gen_time(s) | Old_time(s) | Upd_time(s) | Total_time(s) | Throughput(tokens/s) |
> |--------|------------|----------|----------|----------|----------|------------|------------|
> | **On-policy RL** | | | | | | | |
> | GRPO   | 153 | 828  | 133 | 24 | 87  | 244 | 3474 |
> | DAPO   | 153 | 833  | 307 | 25 | 92  | 424 | 2011 |
> | **Expert-guided RL** | | | | | | | |
> | LUFFY  | 153 | 2902 | 270 | 60 | 230 | 560 | 5306 |
> | QuestA | 510 | 711  | 142 | 31 | 117 | 290 | 2510 |
> | MENTOR | 153 | 1751 | 404 | 48 | 175 | 627 | 2860 |
>
> **MENTOR achieves the highest performance with only moderate and acceptable additional training overhead.** Since different methods generate responses of different lengths, we mainly rely on throughput for a fair comparison.
> * Compared with on-policy RL methods, MENTOR reaches 2860 tokens/s, between GRPO (3474) and DAPO (2011), because DAPO often performs two or three full generation phases to collect enough samples, while MENTOR’s mixed-policy rollouts are more efficient than repeated full generations.
> * For expert-guided methods, LUFFY shows high throughput partly because it mixes in a full offline expert trajectory of about 6k tokens, which increases the number of processed tokens. From the perspective of the algorithmic design, the throughput of LUFFY’s newly generated rollout data should be close to that of GRPO (3474). QuestA concatenates expert segments into the input, creating longer prompts (153 -> 510) that slightly reduce training throughput. Compared with these approaches, MENTOR achieves the highest final performance, and although it relies on expert guidance during the rollout stage, the additional overhead remains acceptable.
>
> ---
>
> >**Q4:** (major) It is somewhat strange to me that the authors use different expert models for different base LLMs, e.g., OpenR1-Qwen-7B model for Qwen models and finetuned LLaMA3.1-8B-Instruct model for LLaMa models. Is there any reason that you do not use a fixed expert model? Why do you use the Instruct model for LLaMa here? It is unclear whether the proposed method would fail if one adopts other LLMs as the base model, e.g., deepseek-R1 as the expert model.
>
> We thank the reviewer for the question. Our entropy-based mixed-policy requires token-level entropy from both models, which can be well-defined under the shared-vocabulary assumption stated in Definition 3.1. In practice, this assumption is easily satisfied, because existing model families (e.g., Qwen2.5) already offer a wide choice of experts with different size (7B–72B) and different domain (e.g., math-centric, code-centric), all under a shared vocabulary. For the expert in LLaMA experiments, we used the version fine-tuned on OpenR1-MATH-220K, rather than the original LLaMA3.1-8B-Instruct.
>
> Although recent work [3] has achieved cross-vocabulary alignment through additional token merging, we use experts from the same model family to conveniently and efficiently validate the effectiveness of our core idea, and leave cross-vocabulary expert guidance as future work.
>
> [3] Unlocking On-Policy Distillation for Any Model Family

---

> ### Author Response · Authors · 2025-11-23
> **Response to Reviewer QY8s (5/6)**
>
> >**Q9:** Why do you still adopt trajectories from the current model in Eq. 9? Any insights here?
>
>
> Retaining the on-policy component in Eq. 9 is essential for **selective absorption** of expert guidance. Under mixed-policy rollouts, the expert often introduces tokens such as “wait”, “verify”, or other reflective markers, and the base model gradually increases the probability of sampling these tokens as it learns from the expert.
>
> However, some tokens are beneficial **only when expert guidance is consistently present**, as the expert can immediately correct deviations and steer the trajectory toward the correct solution. Once expert guidance is removed at inference time, some of these expert tokens may become ineffective or even harmful. For example, “wait” can trigger extended self-reflection loops under pure on-policy rollouts, causing the model to produce unnecessarily long chains that eventually hit the length limit and fail.
>
> The on-policy trajectories therefore serve to **identify which behaviors from expert remain useful when the base model acts alone**. This allows MENTOR to retain expert behaviors that generalize without guidance, while naturally filtering out those that only function under continuous expert supervision.
>
> ---
>
> >**Q10:** In Eq. 11, you write $A\_i=\frac{[R\_i-\text{mean}(R\_j)]\_+}{\text{Range}}$ rather than $A\_i=\frac{[R\_i-\text{mean}(R\_j)]\_+}{\text{Std}}$ , why? Why only normalize the advantages from the mixed policy into [0, 1] but use Eq. 10 directly from GRPO?
>
>
> In Eq. 11, we normalize the advantage of mixed-policy trajectories using
> $
> A\_i=\frac{[R\_i-\overline{R}\_{\text{on}}]\_+}{R\_{\text{range}}}
> $
> rather than dividing by the standard deviation. This choice is deliberate.
>
> For mixed-policy trajectories, the baseline is the **mean reward of the on-policy group** for the same question, which allows us to measure whether expert-guided reasoning truly outperforms the model’s own reasoning. Under this formulation, using the standard deviation would distort this comparison, thereby weakening our ability to assess the expert’s contribution.
>
> This design is essential not only for measuring contribution, but also for ensuring **MENTOR’s robustness to expert quality**. Because the advantage is defined relative to the on-policy baseline, MENTOR remains stable even under extreme conditions—for example, when the expert produces only incorrect answers. In such cases, MENTOR still guarantees a performance lower bound equivalent to GRPO.
>
> This robustness arises from defining the expert-guided advantage, which induces two effects:
> 1. **Rejection of incorrect expert guidance.**
> The term $[R\_i-\overline{R}\_{\text{on}}]\_+$ implicitly filters out all failed expert-guided trajectories by assigning them zero advantage, preventing misleading guidance from influencing learning.
>
> 2. **Selective reinforcement of genuine improvements.**
> Only when the expert-guided trajectory outperforms the model’s own reasoning (i.e., $R_i > \overline{R}_{\text{on}}$) does it receive positive weight. This lets the algorithm identify whether success comes from the model or from the expert and reinforce only true improvements.
>
> Consequently:
>
> * **Any incorrect or unhelpful expert guidance is neutralized.**
> * **Valuable guidance is amplified in proportion to its contribution.**
>
> Thus, this formulation enables MENTOR to remains robust even when the expert is weak, and even if the expert provides only incorrect guidance, it still guarantees a performance lower bound equivalent to GRPO.

---

> ### Author Response · Authors · 2025-11-23
> **Response to Reviewer QY8s (6/6)**
>
> >**Q5:** (major) It is also unclear whether the proposed method still works if the expert model is only marginally stronger than the base model, or if the expert model is too strong (e.g., Qwen3-max), and whether the proposed method can still work. Note that QuestA conducts experiments using some strong base LLMs. The authors commented that the advantages of QuestA rely heavily on the vanilla performance of the base models, while it raises questions on whether MENTOR can work if it is adopted upon some strong base models (e.g., Nemotron-1.5B)
> >
> >**Q11:** Can the proposed MENTOR still work when the base LLM is strong?
>
>
> We thank the reviewer for the insightful question. As suggested, we evaluated MENTOR on a strong base model, Nemotron-1.5B, while keeping OpenR1-Qwen-7B as the expert model.
>
> **Table R6:** Results of MENTOR Applied to a Strong Base Model (Nemotron-1.5B)
> | Model            |  MATH | AIME24 | AIME24 | AMC | Minerva | Olympiad |Avg |
> |--------------------|:-----:|:------:|:------:|:-----:|:------:|:------:|:------:|
> | Nemotron-1.5B | 84.6  | 59.4   | **50.0** | 82.9 | 16.9 |53.7 | 57.9
> | MENTOR-Nemotron-1.5B (435 step) | **86.0**  | **63.3**   | 49.5 | **84.0** | **19.1**| **57.6**| **59.9**
>
> The results in Table R6 show that MENTOR remains effective even when the expert is only marginally stronger than the base model, and the method still achieves clear performance improvements (due to limited computational resources, we trained for only 435 steps). This robustness arises from MENTOR’s ability to selectively extract valuable expert signals, rather than depending on uniformly strong guidance. We discuss this mechanism in detail in Appendix A.1.3 and in our responses to your Q9 and Q10.
>
>
> >**Q6:** (minor) The related work part can be significantly improved. LLM reasoning and LLM reasoning under guidance are fast-growing research areas, and there are numerous papers that do similar things. The authors should include more discussion in the manuscript
>
>
> We thank the reviewer for the suggestion. We will survey the relevant literature in guided LLM reasoning and incorporate additional related work into the next revision of the manuscript.
>
> ---
>
> >**Q7:** (minor) No full pseudo-code for MENTOR can be found. Algorithm 1 just states part of the MENTOR algorithm
>
> We thank the reviewer for the comment. Figure 1 in the manuscript presents the main workflow of MENTOR. We are currently preparing a full pseudo-code version of the algorithm, and we will include it in the corresponding section in the next revision.
>
> ---
>
> >**Q8:** (minor) There are numerous minor issues in the manuscript. I just list some of them below, please double-check your manuscript
>
> Thank you for your careful and thorough review. We have corrected these typos and rechecked the manuscript to ensure the issue is fully resolved.

---

> ### Author Response · Authors · 2025-11-27
> **Completed Revisions Addressing Q6 and Q7**
>
> Dear Reviewer,
> ﻿
>
> We would like to express our sincere gratitude for your constructive insights on our submission, which has been extremely helpful for improving the quality of our work.
> ﻿
>
> **In the latest revision of the manuscript**, we have incorporated **the related work on LLM reasoning and guided reasoning (Q6)** in Section 5 (Related Work) and provided **a pseudocode implementation of MENTOR (Q7)** in Appendix A.2 for your reference. With these updates, we have provided point-by-point responses to all of your questions and hope that our revisions adequately address all of your concerns. We would greatly appreciate it if you could take the time to review our responses.
> ﻿
>
> If there are any additional points that require further clarification, please feel free to let us know. Your insights are invaluable to us, and we're eager to address any remaining concerns to further strengthen our work.
> ﻿
>
> With sincere regards,
>
> The author of 17984: Selective Expert Guidance for Effective and Diverse Exploration in Reinforcement Learning of LLMs

---

> > ### Comment · Reviewer_QY8s · 2025-11-28
> >
> > Thank you for the detailed rebuttal. I have some follow-up:
> >
> > > To strengthen this connection, we extend the analysis in Appendix A.1.1 and introduce a new theoretical result, Theorem 2.1
> >
> > I think adding new theoretical results is usually not encouraged (not sure whether it is allowed) in this venue. Theorem 2.1 is also a bit misleading since it does not explicitly show the upper bound, but concludes with some quantitative analysis on $\Delta H$ with some comparatively strong assumptions. For example, the conclusion may not hold if $c_1 < \dfrac{K}{N} < c_2$
> >
> > > About Eq. 11
> >
> > Have you conducted any experiments to justify your claim? Intuitive ideas sometimes may not work in LLMs
> >
> > > Nemotron-1.5B experiments
> >
> > Why do the reported scores of Nemotron-1.5B seem to be inferior to those reported in the QuestA paper? The performance of MENTOR also seems to lie behind that of QuestA. Can one interpret that MENTOR is inferior to QuestA when the base LLM is strong and can only work better when the base LLM is comparatively weak (and the expert model is strong)? I am also curious about the choice of training steps. 435 steps seem quite like a cherry-picked step with the highest validation score.
> >
> > I would like to hear further clarifications from the authors.

---

> ### Author Response · Authors · 2025-12-03
>
> Dear Reviewer，
>
> Thank you very much for your thoughtful follow-up. We address the new questions you raised point by point below.
>
> > **Q11:** About Theorem 2.1
>
> Thank you for the reviewer’s reminder. We carefully checked the ICLR 2026 Author Guide, and confirm that it does not place restrictions on adding theoretical results during the discussion stage. Moreover, Theorem 2.1 is not an entirely new result—it is a direct application of the conclusions already established in the original Appendix A.1.1, presented in theorem form to improve the organization of the paper.
>
> The entropy upper bound was already provided in the proof of Theorem 2.1, and in the revised manuscript we have made this upper bound explicit in the statement of Theorem 2.1 to avoid any ambiguity.
>
> Regarding the reviewer’s concern about the condition $c_1 < \\tfrac{K}{N} < c_2$:
> In our definition we explicitly assume that $c_1$ is larger than the expected reward under the random (uniform) policy. Notably, $\\tfrac{K}{N}$ is exactly the expected reward of a random policy, as it represents the probability of sampling an optimal trajectory uniformly from the entire exploration space.
>
> Therefore, **the reviewer’s concern would only arise in the extreme case where the pre-trained LLM performs worse than a random policy.** In practice, this is highly unlikely, because pre-trained LLMs already possess substantial prior knowledge and thus easily achieve expected rewards that are higher than the random-policy baseline, even before any RL fine-tuning begins. Consequently, the condition $c_1 > \tfrac{K}{N}$ is easily satisfied in RL settings of LLMs.
>
>
>
>
> > **Q12:** Have you conducted any experiments to justify your claim? Intuitive ideas sometimes may not work in LLMs
>
> We have added the ablation results and analyses of different normalization strategies in Appendix A.5.4. The experiments show that **range-based normalization enables more stable absorption of expert knowledge.**
>
> > **Q13:** Nemotron-1.5B experiments
>
> The reason why Nemotron-1.5B’s reported scores in our experiments are lower than those reported in the QuestA paper is that QuestA’s results were obtained under substantially larger training budgets—for example, over 1k training steps, longer response lengths, larger rollout nums, and dynamic sampling. Due to the limited author discussion period and our computational resources constraints, the latest checkpoint available at the time of our first response was at 435-th step, which is why we used that checkpoint.
>
> To enable a more fair comparison with QuestA, we conducted the QuestA baseline experiments under the same configuration as our main experiments, except that we increased the maximum sequence length to 15,000 (matching the MENTOR setting used in the previous response). We compared performance at both 200 and 400 training steps, and the results are summarized below.
>
>
> **Table R7:** Results of MENTOR and QuestA Applied to a Strong Base Model (Nemotron-1.5B)
> | Model            |  MATH | AIME24 | AIME24 | AMC | Minerva | Olympiad |Avg |
> |--------------------|:-----:|:------:|:------:|:-----:|:------:|:------:|:------:|
> | Nemotron-1.5B | 84.6  | 59.4   | 50.0 | 82.9 | 16.9 | 53.7 | 57.9
> | QuestA-Nemotron-1.5B (200 step) | 85.0  | 60.7   | 50.8 | 83.2 | 17.0| 54.4| 58.5
> | QuestA-Nemotron-1.5B (400 step) | 85.4  | **63.3**   | 50.3 | **84.2** | 16.2| 56.6| 59.3
> | MENTOR-Nemotron-1.5B (200 step) | 85.2  | 61.4   | **51.9** | 83.0 | 17.6| 54.4| 58.9
> | MENTOR-Nemotron-1.5B (400 step) | **85.6**  | 62.7   | 50.4 | 83.8 | **19.8**| **57.8**| **60.0**
>
>
> **These results show that MENTOR is consistently effective regardless of the capability of the base model**. Notably, compared with QuestA, MENTOR delivers clear improvements on Minerva, where the base model is highly weak. This further demonstrates that MENTOR's effectiveness holds even in the tasks where the base model has limited capability.
>
> Finally, we emphasize that existing hint-based methods, from R³[1] to QuestA and others[2,3], lack any substantive methodological differences: they all rely on prepending expert trajectories as prompts to guide reasoning. In contrast, **MENTOR provides a genuinely novel form of guidance that departs from these prior hint-based approaches, which is precisely where its value lies.**
>
> [1] Training Large Language Models for Reasoning through Reverse Curriculum Reinforcement Learning, ICML, 2024 \
> [2] BREAD: Branched Rollouts from Expert Anchors Bridge SFT & RL for Reasoning, NeurIPS, 2025 \
> [3] GHPO: Adaptive Guidance for Stable and Efficient LLM Reinforcement Learning, Arxiv, 2025

---

### Official Review · Reviewer_ugNa · 2025-10-30

**Soundness:** 3
**Presentation:** 3
**Contribution:** 3
**Rating:** 4
**Confidence:** 3

**Summary:**

MENTOR addresses the challenge of balancing effectiveness and diversity in Reinforcement Learning with Verifiable Rewards (RLVR). The method mixes the policy and expert distributions at the token level using entropy-based weights, and introduces an “accelerated mixed-policy rollout” to improve sampling efficiency.
Experiments on Qwen2.5 and LLaMA3.1 show consistent gains across in-domain and out-of-domain reasoning benchmarks.

**Strengths:**

1.1 The motivation is clear and well-grounded. The paper directly tackles the exploration–effectiveness trade-off in RLVR.

1.2 The proposed PMix mechanism (Eq.6) is conceptually interesting, using entropy as a proxy for critical decision points.

1.3 Results are consistently strong across multiple model scales and benchmarks, showing stable gains.

**Weaknesses:**

2.1 The “accelerating mixed-policy rollout” (Eq.7) appears inconsistent. Even with a small wt (e.g., 0.01), the expert π* must still be evaluated to compute the mixed distribution, so the method does not actually avoid the cost of expert inference. A gating mechanism for small weights would be necessary for genuine speed-up.

2.2 The claimed parallelization (L216–220) is unclear. Eq.7 explicitly depends on π*, so it cannot be fully parallelized in the autoregressive decoding process. The efficiency argument based on speculative sampling seems overstated.

2.3 Missing ablations for critical parameters (γp, α) and no exploration of alternative PMix strategies.

2.4 In Appendix A.2, the authors set the KL regularization coefficient to 0 during MENTOR fine-tuning.
This is concerning, as the KL term is typically used to constrain the policy from drifting too far from the base model. Disabling it may weaken stability and allow distributional drift, which the paper does not quantify or justify empirically.

2.5 Results appear to be based on single runs, with no reported variance or standard deviations. This limits statistical reliability and makes it difficult to assess the robustness or significance to the provided results.

**Questions:**

3.1 Can the authors clarify how the “accelerated rollout” avoids computing π* at every step when wt is small?

3.2 Did you test other mixture strategies (e.g., thresholded expert calls or adaptive gating) to validate that entropy weighting is optimal?

3.3 How sensitive are results to γp and α scheduling?

3.4 Can you provide more seeds for the results?

3.5 Could you report results with a non-zero KL coefficient to show how it affects stability and performance?  Also, please clarify the reasoning behind setting the KL term to 0 during MENTOR fine-tuning.

I am willing to increase my score if the authors adress my concerns/questions.

---

> ### Author Response · Authors · 2025-11-23
> **Response to Reviewer ugNa (1/4)**
>
> Thank you for your valuable feedback to help us improve our paper. We have revised our paper based on your feedback. We detail our response below and please kindly let us know if our response addresses your concerns.
>
> ---
>
> > **Q1:** The “accelerating mixed-policy rollout” (Eq.7) appears inconsistent. Even with a small wt (e.g., 0.01), the expert π* must still be evaluated to compute the mixed distribution, so the method does not actually avoid the cost of expert inference. A gating mechanism for small weights would be necessary for genuine speed-up.
> >
> > **Q6:** Can the authors clarify how the “accelerated rollout” avoids computing $\pi^*$ at every step when wt is small?
>
> There seems to be a misunderstanding: **We are not skipping the computation of $\pi^*$.** Our acceleration comes from replacing the token-by-token computation of $\pi^\*$ with computing $\pi^*$ over many tokens at once.
>
> Before providing a more detailed explanation, allow me to first give a brief introduction to the acceleration principle about speculative decoding. The key is that **computing logits for a short continuation of length K in parallel has nearly the same latency as generating a single token**[1]. And this parallelism can translate into actual speed-up when most draft tokens are accepted by the target model. When acceptance is high, the target model pays roughly the cost of producing one token but obtains several verified tokens in a single pass.
>
> Our accelerated mixed-policy rollout explicitly leverages this principle. The key observation is that **$\pi_\text{mix}$ differs from the base policy $\pi_\theta$ only at a sparse set of high-entropy positions, while for the vast majority of tokens we have $w_t \approx 0$ and thus $\pi_\text{mix}\approx\pi_\theta$.** When $\pi_\theta$ is used as the draft model, this means that most draft tokens match the mixed-policy distribution closely enough to be accepted almost always. With such a high acceptance rate, each speculative generation step allows the expert to validate K draft tokens from $\pi_\theta$ within a single forward pass, and then accept the most of them, thereby substantially accelerating mixed-policy sampling.
>
> Regarding the discrete gating mechanism, **although gating removes the need for $\pi^*$ when constructing $\pi_{\text{mix}}$ at certain positions, it cannot eliminate the expert’s computational cost at those positions.** LLMs must first prefill the entire prefix before generating the next token. Therefore, even if a specific token $y_t$ is sampled at a step where the gate is closed, as soon as we later turn the gate open at some step $t' > t$, the expert model $\pi^*(\cdot \mid q, y_{<t'})$ must still be computed on the entire prefix, which necessarily includes all earlier tokens $(y_1,\dots,y_t)$.  And our method precisely leverages this prefill to validate draft tokens in parallel, allowing expert guidance to permeate every token and provide richer learning signals, while introducing almost no additional overhead compared to discrete gating.
>
> In summary, **the acceleration does not come from skipping the expert, but from letting the expert validate many tokens at once rather than one token.***
>
> ---
>
> >**Q2:** The claimed parallelization (L216–220) is unclear. Eq.7 explicitly depends on $\pi^*$, so it cannot be fully parallelized in the autoregressive decoding process. The efficiency argument based on speculative sampling seems overstated.
>
> Let me clarify the misunderstanding about parallelization. The parallelization we refer to is that the draft model produces K tokens sequentially, but the expert verifies all K tokens **in parallel** by a single forward pass.
>
>
> [1] Accelerating Large Language Model Decoding with Speculative Sampling

---

> ### Author Response · Authors · 2025-11-23
> **Response to Reviewer ugNa (2/4)**
>
> >**Q3:** Missing ablations for critical parameters (γp, α) and no exploration of alternative PMix strategies.
> >
> > **Q7:** Did you test other mixture strategies (e.g., thresholded expert calls or adaptive gating) to validate that entropy weighting is optimal?
> >
> > **Q8:** How sensitive are results to γp and α scheduling?
>
>
> We thank the reviewer for pointing this out. To address these concerns, we have conducted a comprehensive set of ablation studies covering
> * (i) MENTOR’s core components,
> * (ii) key hyperparameters—including the expert weight $\alpha$ and the entropy threshold $\gamma_p$
> * (iii) alternative mixing strategies.
>
> All results are now included in Appendix A.3–A.4 of the revised paper.
>
> ### **Ablation on components of MENTOR**
> **Table R1:** Main results of **component ablation** of MENTOR
> | Method                 | MATH | AIME24 |
> |------------------------|:----:|:------:|
> | Qwen2.5-7B-Base        | 62.4 |  5.4   |
> | GRPO                   | 76.8 | 14.2   |
> | +Mixed-policy Rollout  | 79.4 | 14.6   |
> | +Mixed-policy GRPO     | **81.4** | **18.3** |
>
> **Each component contributes.** The observed improvements demonstrate the effectiveness of these components in RL training, with each contributing performance gains on MATH.
>
> ### **Ablation on the Expert weight $\alpha$**
> **Table R2:** Ablation on the **Expert weight $\alpha$** in MENTOR
> | Setting                       | MATH | AIME24 |
> |------------------------------|:----:|:------:|
> | GRPO (equiv. to $\alpha$ = 0)       | 76.8 | 14.2   |
> | MENTOR (fixed $\alpha$ = 1.0)       | 78.2 | 13.9   |
> | MENTOR (fixed $\alpha$ = 0.5)       | 80.4 | 16.1   |
> | MENTOR (decay $\alpha$ : 1 → 0)     | **81.4** | **18.3** |
>
> We identified the following findings in Ablation on the Expert weight $\alpha$ :
> 1. **Introducing expert knowledge consistently improves performance.**
> Across all settings, MENTOR outperforms GRPO ($\alpha = 0$ ), showing that adding expert information reliably boosts results.
>
> 2. **Moderate expert weighting outperforms an excessively high weight.**
> Although both $\alpha = 0.5$ and $\alpha = 1$ use expert guidance, the former achieves higher scores, indicating that moderate mixing, which preserves some degree of autonomous exploration, leads to better performance than excessively relying on expert guidances.
>
> 3. **A decaying schedule delivers the best results.**
> The decaying schedule achieves the highest performance, suggesting that gradually reducing the expert weight enables a smooth shift from guided learning to autonomous optimization.
>
> ### **Ablation on the Entropy Threshold $\gamma_p$**
> **Table R3:** Ablation on the **Entropy Threshold** $\gamma_p$ in MENTOR
> | Setting            |  MATH | AIME24 |
> |--------------------|:-----:|:------:|
> | MENTOR *(p = 0.8)* | 80.8  | 17.0   |
> | MENTOR *(p = 0.9)* | 80.2  | 17.7   |
> | MENTOR *(p = 0.95)*| **81.4** | **18.3** |
>
> **MENTOR’s final performance remains stable across different $\gamma_p$.** Table R3 shows that the final results vary only slightly with the threshold, demonstrating that MENTOR is robust to the choice of $\gamma_p$.

---

> ### Author Response · Authors · 2025-11-23
> **Response to Reviewer ugNa (3/4)**
>
> ### **Exploring Alternative Forms of Mixed-policy**
> We adopt entropy as the selection principle because prior studies[2][3] show that high-entropy tokens typically are critical decision points in the reasoning process.
>
> We also agree with the reviewers that examining alternative selection strategies is important for assessing whether entropy is indeed the most effective choice compared with other options. We further investigate several alternative ways, including random/uniform guidance (both are equivalent) and based-perplexity guidance. We have revised the paper and put these details in Appendix A.3.
>
> 1. **Random/uniform guidance.**
> This baseline injects expert tokens uniformly at random with a fixed probability, yielding the mixed policy
> $$
> \pi_{\text{mix}}(y_t \mid x_{<t}) = 0.8\pi_\theta(y_t \mid x_{<t}) + 0.2\pi^{*}(y_t \mid x_{<t}).
> $$
>
> 2. **Perplexity-based guidance.**
> This strategy routes tokens with perplexity above the 80th-percentile threshold $\tau$ to the expert, producing the policy
> $$
> \pi_{\text{mix}}(y_t \mid x_{<t})=
> \begin{cases}
> \pi^{*}(y_t \mid x_{<t}) & \mathrm{PPL}(t) > \tau,\\
> \pi_{\theta}(y_t \mid x_{<t}) & \text{otherwise}.
> \end{cases}
> $$
>
> To provide a direct illustration of how these guidance mechanisms differ in practice, we further visualize the critical tokens generated by the expert in Appendix A.3.
>
> We identified the following findings:
>
> 1. **Entropy-based guidance frequently introduces logical connectors** (e.g., *wait*, *however*), which often trigger new reasoning branches and produce trajectories whose style and structure deviate significantly from the model’s unguided reasoning.
> 2. **Random and perplexity-based guidance seldom create such branching points**, yielding trajectories that stay close to those generated by the base model.
> 3. **Entropy-based guidance yields the largest performance gains**, substantially outperforming random and perplexity-based guidance on both MATH and AIME24 as shown in Table R4, indicating that expert guidance is most effective when applied at high-entropy positions.
>
>
> **Table R4:** Impact of **different guidance** on MENTOR performance.
> | Setting                               | MATH | AIME24 |
> |----------------------------------------|:----:|:------:|
> | GRPO                                  | 76.8 | 14.2   |
> | MENTOR (Random guidance)              | 77.6 | 14.8   |
> | MENTOR (Perplexity-based guidance)    | 77.0 | 13.3   |
> | MENTOR (Entropy-based guidance)       | **81.4** | **18.3** |
>
> [2] Beyond the 80/20 Rule: High-Entropy Minority Tokens Drive Effective Reinforcement Learning for LLM Reasoning \
> [3] Blending Supervised and Reinforcement Fine-Tuning with Prefix Sampling

---

> ### Author Response · Authors · 2025-11-23
> **Response to Reviewer ugNa (4/4)**
>
> >**Q4:** In Appendix A.2, the authors set the KL regularization coefficient to 0 during MENTOR fine-tuning. This is concerning, as the KL term is typically used to constrain the policy from drifting too far from the base model. Disabling it may weaken stability and allow distributional drift, which the paper does not quantify or justify empirically.
> >
> >**Q10:** Could you report results with a non-zero KL coefficient to show how it affects stability and performance? Also, please clarify the reasoning behind setting the KL term to 0 during MENTOR fine-tuning.
>
> We appreciate the reviewer’s concern. But removing the KL penalty is actually a common practice in long-CoT reinforcement learning, first introduced by **DAPO**[4] and **Dr.GRPO**[5].
>
> The KL term is designed to constrain the online policy from drifting too far from a frozen reference policy. This constraint is crucial in RLHF settings, where the objective is alignment and the model is not expected to deviate substantially from the base distribution. However, for **long-CoT reasoning**, the situation is fundamentally different: the optimal solution paths often lie far outside the base model’s distribution. Forcing the policy to stay close to the initial model can overly restrict exploration. Therefore, many subsequent reasoning-oriented methods, including our baselines **LUFFY** and **QuestA**, also remove the KL term during RL training.
>
> We understand the reviewer’s concern about potential distributional drift. To verify robustness, we followed the main-experiment setup and conducted an additional experiment on **Qwen2.5-7B-Base** with a non-zero KL coefficient ($\beta$ = 0.04). The results show that:
>
> * **Without KL**, the model reaches its peak accuracy on **MATH500** at *150 steps* with **0.768**.
> * **With KL**, the peak accuracy appears later, at *195 steps*, and is slightly lower (**0.762**).
>
> These results indicate that disabling KL leads to faster improvement and slightly better final accuracy in our setting.
>
> >**Q5:** Results appear to be based on single runs, with no reported variance or standard deviations. This limits statistical reliability and makes it difficult to assess the robustness or significance to the provided results.
> >
> >**Q9:** Can you provide more seeds for the results?
>
> Due to time constraints, we could not rerun all experiments with multiple seeds. Therefore, we repeated the key comparisons on Qwen2.5-3B-Base for both GRPO and MENTOR under **three random seeds: 1 (default), 42, and 2025**. We now report the mean and standard deviation across these runs for all evaluation metrics.
>
> **Table R5:** Results from Repeated Experiments Across Multiple Random Seeds(1, 42, 2025)
> | Methods       | MATH        | AIME24      | AIME25      | AMC         | Minerva     | Olympiad    | GPQA        | ARC         | MMLU-Pro    | Avg         |
> |---------------|:-----------:|:-----------:|:-----------:|:-----------:|:-----------:|:-----------:|:-----------:|:-----------:|:-----------:|:-----------:|
> | Qwen2.5-3B-Base          | 47.4   | 2.4    | 1.9    | 17.7   | 19.9   | 19.0   | 3.0    | 23.6   | 19.4   | 17.1   |
> | On-policy RL  | 65.7 ± 1.3  | 3.6 ± 0.4   | 2.5 ± 0.3   | 32.5 ± 1.2  | 25.6 ± 0.7  | 31.2 ± 2.6  | 17.9 ± 2.3  | 74.6 ± 3.2  | 31.6 ± 1.4  | 31.7 ± 0.7  |
> | MENTOR        | **70.0** ± 0.9  | **8.6** ± 0.7   | **4.5** ± 0.5   | **34.5** ± 0.5  | **26.6** ± 1.3  | **34.7** ± 1.1  | **21.1** ± 1.7  | **80.7** ± 1.8  | **34.8** ± 2.8  | **35.1** ± 0.2  |
>
> Across all metrics, MENTOR consistently outperforms on-policy RL, demonstrating that its improvements are stable, reproducible, and statistically reliable.
>
>
>
>
> [1] Accelerating Large Language Model Decoding with Speculative Sampling \
> [2] Beyond the 80/20 Rule: High-Entropy Minority Tokens Drive Effective Reinforcement Learning for LLM Reasoning \
> [3] Blending Supervised and Reinforcement Fine-Tuning with Prefix Sampling \
> [4] DAPO: An Open-Source LLM Reinforcement Learning System at Scale \
> [5] Understanding R1-Zero-Like Training: A Critical Perspective

---

> > ### Comment · Reviewer_ugNa · 2025-11-23
> >
> > Thank you for yourdetailed explanation with the extra results, therefore I am increasing my score.

---

> > > ### Author Response · Authors · 2025-11-24
> > > **Thanks for your time and thoughtful feedback!**
> > >
> > > Thank you for your feedback and for increasing your score! Your thoughtful review and support are greatly appreciated. If you have any further questions, please feel free to contact us!

---

### Official Review · Reviewer_Kxjd · 2025-10-31

**Soundness:** 3
**Presentation:** 3
**Contribution:** 2
**Rating:** 4
**Confidence:** 3

**Summary:**

This paper introduces MENTOR (Mixed-policy Expert Navigation for Token-level Optimization of Reasoning), a GRPO+ training method for improving the reasoning capabilities. By providing guidance on critical tokens during inference, MENTOR achieves effective improvement of reasoning while preserving the diversity of group-based sampling. Experiments on math and science benchmarks show that MENTOR consistently outperforms over baselines like GRPO, LUFFY, and QuestA.

**Strengths:**

1. The motivation of the paper is reasonable, while full expert guidance could lead to the reduction of sampling diversity, MENTOR implements a relaxed guidance that guides the sampling only on critical tokens, leading to the balance of training efficacy and sampling diversity.

2. The MENTOR also involves an accelerated components that address the efficiency problem of the expert-guidance method, which is also a unique contribution.

3. This paper does comprehensive experiments on both in-domain and out-of-domain benchmarks, showing the efficacy and generalizability of MENTOR. In addition, the analysis results of entropy show that MENTOR can preserve the generation diversity during training.

**Weaknesses:**

1. One of the main weaknesses is that the mechanism of "MENTOR only guides the decoding on critical tokens". This paper shows MENTOR's decoding strategy in Sec. 3.1, but does not compare this strategy with prior methods of fully expert-guided decoding. A more detailed introduction and explanation of the decoding strategy is required.

2. The motivation for using an expert-guided method is not entirely clear. If a powerful expert model is already available, it seems straightforward to use it directly for reasoning tasks. Why is it necessary to further train and distill its knowledge into a smaller model? I would appreciate a more detailed discussion on the advantages and rationale behind training a smaller model with expert guidance.

3. The expert-guided method costs more training resources compared with vanilla GRPO, since it requires loading a bigger expert model for decoding, which will occupy a large amount of GPU resources. Even though this paper proposes an accelerated strategy, the extra cost of GPU computation and memory can not be overlooked.

**Questions:**

1. Could the authors provide a more detailed comparison between MENTOR and DAPO? Although DAPO may require additional time for resampling, it does not incur the extra GPU memory overhead associated with loading a separate expert model. In this respect, DAPO could be a more resource-efficient alternative to MENTOR.

2. Could the authors provide concrete case studies to better illustrate MENTOR's impact on sampling diversity? While Figure 3 presents keyword statistics, a more direct qualitative analysis or examples would offer clearer insight into the diversity achieved through MENTOR.

3. Could the authors provide an explanation about changing the selected expert model rather than OpenR1-Qwen-7B? How does performance change if the expert is weaker or from a different architecture/family?

---

> ### Author Response · Authors · 2025-11-23
> **Response to Reviewer Kxjd (1/2)**
>
> Thank you for your valuable feedback to help us improve our paper. We have revised our paper based on your feedback. We detail our response below and please kindly let us know if our response addresses your concerns.
>
> > **Q1:** One of the main weaknesses is that the mechanism of "MENTOR only guides the decoding on critical tokens". This paper shows MENTOR's decoding strategy in Sec. 3.1, but does not compare this strategy with prior methods of fully expert-guided decoding. A more detailed introduction and explanation of the decoding strategy is required.
>
> Thank you for your valuable suggestion. We have revised the paper and put more details about fully expert-guided decoding in Sec. 3.1.
>
> > **Q2:** The motivation for using an expert-guided method is not entirely clear. If a powerful expert model is already available, it seems straightforward to use it directly for reasoning tasks. Why is it necessary to further train and distill its knowledge into a smaller model? I would appreciate a more detailed discussion on the advantages and rationale behind training a smaller model with expert guidance.
>
> Although expert models achieve stronger performance, they usually have far more parameters, making deployment and updates expensive. Smaller models, in contrast, are easier to deploy locally and support continual training. However, because pure on-policy RL cannot effectively expand the model’s capability frontier[1], smaller models often end up with weaker final performance. This motivates our work, which uses expert guidance to help small models explore more effectively and achieve higher performance. We will incorporate this clarification and update the manuscript accordingly.
>
> >**Q3:** The expert-guided method costs more training resources compared with vanilla GRPO, since it requires loading a bigger expert model for decoding, which will occupy a large amount of GPU resources. Even though this paper proposes an accelerated strategy, the extra cost of GPU computation and memory can not be overlooked.
>
> We appreciate the reviewer’s concern regarding the additional computational cost introduced by expert-guided approaches. To address this point, we conduct a dedicated efficiency analysis during 200 training steps on Qwen2.5-7B-Base (see Appendix.A.5 for details). Additionally, because different RL methods produce responses of substantially different lengths, we further define an **throughput** metric to ensure fair comparison across methods, which is computed as the average number of tokens that produce gradients per step divided by the average per-step time. All results are summarized in Table R1
>
> **Table R1:** Efficiency analysis of different methods. Here, Gen, Old and Update denote respectively the generation (rollout) phase, the computing of the logits of $\pi_\text{old}$, and the model update phase in the Verl framework.
> | Method | Prompt_len | Resp_len | Gen_time(s) | Old_time(s) | Upd_time(s) | Total_time(s) | Throughput(tokens/s) |
> |--------|------------|----------|----------|----------|----------|------------|------------|
> | **On-policy RL** | | | | | | | |
> | GRPO   | 153 | 828  | 133 | 24 | 87  | 244 | 3474 |
> | DAPO   | 153 | 833  | 307 | 25 | 92  | 424 | 2011 |
> | **Expert-guided RL** | | | | | | | |
> | LUFFY  | 153 | 2902 | 270 | 60 | 230 | 560 | 5306 |
> | QuestA | 510 | 711  | 142 | 31 | 117 | 290 | 2510 |
> | MENTOR | 153 | 1751 | 404 | 48 | 175 | 627 | 2860 |
>
> **MENTOR achieves the highest performance with only moderate and acceptable additional training overhead.** Since different methods generate responses of different lengths, we mainly rely on throughput for a fair comparison.
> * Compared with on-policy RL methods, MENTOR reaches 2860 tokens/s, between GRPO (3474) and DAPO (2011), because DAPO often performs two or three full generation phases to collect enough samples, while MENTOR’s mixed-policy rollouts are more efficient than repeated full generations.
> * For expert-guided methods, LUFFY shows high throughput partly because it mixes in a full offline expert trajectory of about 6k tokens, which increases the number of processed tokens. From the perspective of the algorithmic design, the throughput of LUFFY’s newly generated rollout data should be close to that of GRPO (3474). QuestA concatenates expert segments into the input, creating longer prompts (153 -> 510) that slightly reduce training throughput. Compared with these approaches, MENTOR achieves the highest final performance, and although it relies on expert guidance during the rollout stage, the additional overhead remains acceptable.
>
>
> [1] Does Reinforcement Learning Really Incentivize Reasoning Capacity in LLMs Beyond the Base Model?

---

> ### Author Response · Authors · 2025-11-23
> **Response to Reviewer Kxjd (2/2)**
>
> >**Q4:** Could the authors provide a more detailed comparison between MENTOR and DAPO? Although DAPO may require additional time for resampling, it does not incur the extra GPU memory overhead associated with loading a separate expert model. In this respect, DAPO could be a more resource-efficient alternative to MENTOR.
>
> We provide a detailed comparison between MENTOR and DAPO from the following two aspects.
> 1. **Efficiency analysis.** As shown in Table R1, MENTOR reaches 2860 tokens/s, between GRPO (3474) and DAPO (2011), because DAPO often performs two or three full generation phases to collect enough samples, especially at the early stage with low accuracy, while MENTOR’s mixed-policy rollouts are more efficient than repeated full generations.
> 2. **Capability-boundary expansion.** Capability-boundary expansion. Recent studies[1] show that pure on-policy RL cannot expand a model’s capability boundary (pass@k). DAPO, as an on-policy method, inherits this limitation and therefore cannot improve the model on problems beyond its existing capability range—it merely skips them. In contrast, MENTOR leverages expert guidance to enable the model to explore reasoning trajectories that lie beyond its own capability, thereby overcoming the limitation of pure on-policy RL and expanding the model’s capability boundary, as shown in Figure 4.
>
>
> >**Q5:** Could the authors provide concrete case studies to better illustrate MENTOR's impact on sampling diversity? While Figure 3 presents keyword statistics, a more direct qualitative analysis or examples would offer clearer insight into the diversity achieved through MENTOR.
>
> We thank the reviewer for the constructive suggestion. Besides the keyword statistics presented in Figure 3, our paper also analyzes exploration diversity from two additional perspectives.
> 1. **Generation entropy.**
> Generation entropy directly quantifies the diversity of the model’s output distribution. As analyzed in Sec. 4.3, MENTOR consistently maintains higher entropy throughout training compared with pure on-policy RL. This indicates that MENTOR encourages the model to explore a broader space of reasoning trajectories, whereas pure on-policy RL gradually collapses to low-entropy and low-diversity behaviors.
>
> 2. **pass@k.**
> pass@k is widely used to measure reasoning diversity. As discussed in Sec. 4.5, the improvement in pass@k achieved by MENTOR directly reflects the increased diversity of reasoning trajectories explored during training. Because the model can only learn new solution paths after first discovering them, the improved pass@k reflects MENTOR’s enhanced exploration during training. In contrast, pure on-policy RL shows little improvement—or even slight degradation, indicating that it explores fewer new trajectories and thus fails to enrich the diversity of reasoning.
>
> We will reorganize and integrate these analyses in the revised version to more clearly highlight how MENTOR enhances sampling diversity.
>
> >**Q6:** Could the authors provide an explanation about changing the selected expert model rather than OpenR1-Qwen-7B? How does performance change if the expert is weaker or from a different architecture/family?
>
> We thank the reviewer for the question. Our entropy-based mixed-policy requires token-level entropy from both models, which can be well-defined under the shared-vocabulary assumption stated in Definition 3.1. In practice, this assumption is easily satisfied, because existing model families (e.g., Qwen2.5) already offer a wide choice of experts with different size (7B–72B) and different domain (e.g., math-centric, code-centric), all under a shared vocabulary. Although recent work [2] has achieved cross-vocabulary alignment through additional token merging, we use experts from the same model family to conveniently and efficiently validate the effectiveness of our core idea, and leave cross-vocabulary expert guidance as future work.
>
>
>
> [1] Does Reinforcement Learning Really Incentivize Reasoning Capacity in LLMs Beyond the Base Model? \
> [2] Unlocking On-Policy Distillation for Any Model Family

---

### Official Review · Reviewer_uaif · 2025-10-31

**Soundness:** 2
**Presentation:** 3
**Contribution:** 3
**Rating:** 4
**Confidence:** 3

**Summary:**

This paper introduces MENTOR, a reinforcement learning framework that aims to enhance both the effectiveness and diversity of exploration in RLVR for LLMs. Inject expert guidance only at critical decision points (high-entropy tokens) rather than imitating entire expert trajectories (such as LUFFY). In-Domain and OOD performance evaluation (LLaMa3.1-8B-Base, Qwen2.5-3B-Base and Qwen2.5-7B-base) demonstrate MENTOR's effectiveness.

**Strengths:**

- Clear problem framing: Rollout's effectiveness and diversity are important for RL. Paper's section details description this aspect.
- Critical point guidance: By leveraging token entropy, providing expert guidance only at critical decision points, steers reasoning trajectories while preserving the policy’s own exploration, rather than entirely imitating expert.
- Empirical Results: Based on three backbone models, proposed method MENTOR compared to Vanilla GRPO, LUFFY and QuestA, demonstrate the method's effectiveness.

**Weaknesses:**

- Experiment: The Table 1's experiment appear insufficient, given that the paper’s pipeline involves multiple combinations of hyperparameters and strategies.
- Critical point selection: The selection strategy for critical decision points could be elaborated in more detail — for example, why is it based on token-level entropy rather than other metrics (e.g., higher-level semantic)?
- Typos: Line 322 QuestA appear twice.

**Questions:**

What would happen if the token points for guidance were selected randomly or uniformly?

How robust is the proposed method to the expert’s capability — for instance, if the expert is weaker, can the model filter out misleading guidance?

In the baselines, LUFFY and QuestA belong to expert imitation. Are there any more dynamic or flexible expert guidance baselines for comparison?

If the authors can provide more detailed verification of the method’s effectiveness and robustness, I will increase my score.

---

> ### Author Response · Authors · 2025-11-23
> **Response to Reviewer uaif (1/3)**
>
> Thank you for your valuable feedback to help us improve our paper. We have revised our paper based on your feedback. We detail our response below and please kindly let us know if our response addresses your concerns.
>
> > **Q1:** Experiment: The Table 1's experiment appear insufficient, given that the paper’s pipeline involves multiple combinations of hyperparameters and strategies.
>
>
> To address this, we conducted a series of ablation experiments, including:
> 1. **Component ablation.** We isolate and remove each component of MENTOR to evaluate its individual contribution to overall performance.
> 2. **Hyperparameter ablation.** We analyze the sensitivity of key hyperparameters and quantify their impact on final performance.
>
> We have revised the paper and put these details in Appendix A.4.
>
> **Table R1:** Main results of **component ablation** of MENTOR
> | Method                 | MATH | AIME24 |
> |------------------------|:----:|:------:|
> | Qwen2.5-7B-Base        | 62.4 |  5.4   |
> | GRPO                   | 76.8 | 14.2   |
> | +Mixed-policy Rollout  | 79.4 | 14.6   |
> | +Mixed-policy GRPO     | **81.4** | **18.3** |
>
> **Each component contributes.** The observed improvements demonstrate the effectiveness of these components in RL training, with each contributing performance gains on MATH.
>
> **Table R2:** Ablation on the **Expert weight $\alpha$** in MENTOR
> | Setting                       | MATH | AIME24 |
> |------------------------------|:----:|:------:|
> | GRPO (equiv. to $\alpha$ = 0)       | 76.8 | 14.2   |
> | MENTOR (fixed $\alpha$ = 1.0)       | 78.2 | 13.9   |
> | MENTOR (fixed $\alpha$ = 0.5)       | 80.4 | 16.1   |
> | MENTOR (decay $\alpha$ : 1 → 0)     | **81.4** | **18.3** |
>
> We identified the following findings in Ablation on the Expert weight $\alpha$:
> 1. **Introducing expert knowledge consistently improves performance.**
> Across all settings, MENTOR outperforms GRPO ($\alpha = 0$), showing that adding expert information reliably boosts results.
>
> 2. **Moderate expert weighting outperforms an excessively high weight.**
> Although both $\alpha = 0.5$ and $\alpha = 1$ use expert guidance, the former achieves higher scores, indicating that moderate mixing, which preserves some degree of autonomous exploration, leads to better performance than excessively relying on expert guidances.
>
> 3. **A decaying schedule delivers the best results.**
> The decaying schedule achieves the highest performance, suggesting that gradually reducing the expert weight enables a smooth shift from guided learning to autonomous optimization.
>
> **Table R3:** Ablation on the **Entropy Threshold** $\gamma_p$ in MENTOR
> | Setting            |  MATH | AIME24 |
> |--------------------|:-----:|:------:|
> | MENTOR *(p = 0.8)* | 80.8  | 17.0   |
> | MENTOR *(p = 0.9)* | 80.2  | 17.7   |
> | MENTOR *(p = 0.95)*| **81.4** | **18.3** |
>
> **MENTOR’s final performance remains stable across different $\gamma_p$.** Table R3 shows that the final results vary only slightly with the threshold, demonstrating that MENTOR is robust to the choice of $\gamma_p$.

---

> ### Author Response · Authors · 2025-11-23
> **Response to Reviewer uaif (2/3)**
>
> > **Q2:** Critical point selection: The selection strategy for critical decision points could be elaborated in more detail — for example, why is it based on token-level entropy rather than other metrics (e.g., higher-level semantic)?
> >
> > **Q4:** What would happen if the token points for guidance were selected randomly or uniformly?
>
>
> We adopt entropy as the selection principle because prior studies[1][2] show that high-entropy tokens typically are critical decision points in the reasoning process.
>
> We also agree with the reviewers that examining alternative selection strategies is important for assessing whether entropy is indeed the most effective choice compared with other options. We further investigate several alternative ways, including random/uniform guidance (both are equivalent) and based-perplexity guidance. We have revised the paper and put these details in Appendix A.3.
>
> 1. **Random/uniform guidance.**
> This baseline injects expert tokens uniformly at random with a fixed probability, yielding the mixed policy
> $$
> \pi_{\text{mix}}(y_t \mid x_{<t}) = 0.8\pi_\theta(y_t \mid x_{<t}) + 0.2\pi^{*}(y_t \mid x_{<t}).
> $$
>
> 2. **Perplexity-based guidance.**
> This strategy routes tokens with perplexity above the 80th-percentile threshold $\tau$ to the expert, producing the policy
> $$
> \pi_{\text{mix}}(y_t \mid x_{<t})=
> \begin{cases}
> \pi^{*}(y_t \mid x_{<t}) & \mathrm{PPL}(t) > \tau,\\
> \pi_{\theta}(y_t \mid x_{<t}) & \text{otherwise}.
> \end{cases}
> $$
>
> To provide a direct illustration of how these guidance mechanisms differ in practice, **we further visualize the critical tokens generated by the expert in Appendix A.3.**
>
> We identified the following findings:
>
> 1. **Entropy-based guidance frequently introduces logical connectors** (e.g., *wait*, *however*), as shown in Figure 5 of Appendix A.3, which often trigger new reasoning branches and produce trajectories whose style and structure deviate significantly from the model’s unguided reasoning.
> 2. **Random and perplexity-based guidance seldom create such branching points**, yielding trajectories that stay close to those generated by the base model.
> 3. **Entropy-based guidance yields the largest performance gains**, substantially outperforming random and perplexity-based guidance on both MATH and AIME24 as shown in Table R4, indicating that expert guidance is most effective when applied at high-entropy positions.
>
>
> **Table R4:** Impact of **different guidance** on MENTOR performance.
> | Setting                               | MATH | AIME24 |
> |----------------------------------------|:----:|:------:|
> | GRPO                                  | 76.8 | 14.2   |
> | MENTOR (Random guidance)              | 77.6 | 14.8   |
> | MENTOR (Perplexity-based guidance)    | 77.0 | 13.3   |
> | MENTOR (Entropy-based guidance)       | **81.4** | **18.3** |
>
>
> > **Q3:** Typos: Line 322 QuestA appear twice.
>
> Thank you for your careful and thorough review. We have corrected these typos and rechecked the manuscript to ensure the issue is fully resolved.
>
>
>
> [1] Beyond the 80/20 Rule: High-Entropy Minority Tokens Drive Effective Reinforcement Learning for LLM Reasoning \
> [2] Blending Supervised and Reinforcement Fine-Tuning with Prefix Sampling \

---

> ### Author Response · Authors · 2025-11-23
> **Response to Reviewer uaif (3/3)**
>
> > **Q5:** How robust is the proposed method to the expert’s capability — for instance, if the expert is weaker, can the model filter out misleading guidance?
>
>
> Our method remains robust even when the expert is weak, and even if the expert provides only incorrect guidance, **it still guarantees a performance lower bound equivalent to GRPO.** We have revised the paper and put these details in Appendix A.1.3.
>
> This is primarily because, in Mixed-policy GRPO, we define the advantage of the expert-guided trajectory as $\frac{[R_i - \bar R\_\text{on}]\_+}{R\_\text{range}}$, which creates two effects:
>
> 1. $[R_i - \bar R_\text{on}]_+$, which effectively performs rejection sampling on expert-guided trajectories: any trajectory yielding an incorrect answer is assigned zero advantage, ensuring misleading guidance does not affect learning.
> 2. Only expert-guided trajectories that outperform the model’s own reasoning (i.e., $R(\tau)-\bar{R}\_\text{on}>0$) are reinforced. This allows the algorithm to distinguish whether success comes from the model itself or from the expert guidane, and to assign larger weight to guidance that provides genuine improvements.
>
>
>
> As a result:
>
> * **Any failed or misleading expert guidance is completely filtered out.**
> * **The algorithm adaptively measures the expert’s contribution and assigns larger weight to valuable guidance.**
>
> Thus, the method remains stable and reliable even when the expert is imperfect or weak.
>
> > **Q6:** In the baselines, LUFFY and QuestA belong to expert imitation. Are there any more dynamic or flexible expert guidance baselines for comparison?
>
> In existing RL works, training typically falls into two modes: (i) the model explores entirely on its own without any expert guidance, or (ii) the expert solution trajectory is provided for the model to imitate or use as a reference, as in LUFFY and QuestA. To the best of our knowledge, there is currently no established baseline in RLVR that supports dynamic or selective expert guidance. This absence of a more flexible expert-guided paradigm is precisely the problem our work aims to address.

---

> > ### Comment · Reviewer_uaif · 2025-11-24
> > **Response to Rebuttal**
> >
> > Thank you for your detailed response. I will increase my score.

---

> > > ### Author Response · Authors · 2025-11-24
> > > **Thanks for your time and positive feedback!**
> > >
> > > Thank you very much for your positive feedback and for generously raising your score. We truly appreciate your constructive comments and your engagement with our work. If you have any further questions, please feel free to contact us.

---

### Author Response · Authors · 2025-12-03
**Final Response**

We sincerely thank the new Area Chair for evaluating our submission under the unexpected circumstances caused by the recent OpenReview incident. We also appreciate the original Area Chair and the reviewers for their careful reading and constructive feedback. We understand the additional workload created by the situation and are grateful for the opportunity to provide this final response.

**A.** First, we thank all reviewers for their positive assessments. The main strengths acknowledged across reviews are:

1. **The motivation is clear and well-grounded**. This paper focuses on a key issue in LLM reinforcement learning—the trade-off between exploration and effectiveness. It proposes selective expert guidance to mitigate the diversity collapse caused by full supervision. (Reviewers uaif, Kxjd, ugNa, QY8s)

2. **The core idea is novel and interesting.** MENTOR offers expert guidance only at critical points through a mixed-policy mechanism, and this selective guidance preserves the model’s exploration while improving reasoning effectiveness. (Reviewers uaif, Kxjd, ugNa)

3. **The empirical evaluation is comprehensive.** experiments across multiple model families (Qwen and Llama) and scales, together with comparisons against strong baselines, demonstrate the effectiveness and generality of the proposed method. (Reviewers uaif, Kxjd, ugNa, QY8s)

4. **The paper is clearly written.** The problem definition is clear, and the paper is easy to read and follow, with sufficient technical detail to ensure transparency and credibility. (Reviewers uaif, QY8s)

5. **A unique contribution on efficiency**: The MENTOR also involves an accelerated components that address the efficiency problem of the expert-guidance method. (Reviewer Kxjd)


**B.** We acknowledge the reviewers' concerns, which we have addressed in detail in the reviewer-specific Official Comments. Below, we provide a concise summary of each reviewer's main concerns, our responses, and the reviewers' re-evaluation. Additionally, we have uploaded an updated version of the paper, with all modifications highlighted in dark blue for easy reference.

| Reviewers | Initial Score | Main Concerns / Request | Our Response | Re-evaluated Score |
|----------|---------------|--------------------------|--------------|----------------|
| uaif | 4 | 1. Evaluate additional critical-point selection strategies. (Q2, Q4) | Added more experiments in Appendix A.4 | 6 |
|||2. Robustness of MENTOR to the capability of the expert model (Q5)|We provide further analysis in Appendix A.1.3, which shows that even in extreme cases, MENTOR still guarantees a performance lower bound equivalent to GRPO. ||
| Kxjd | 4 | 1. Efficiency Analysis of MENTOR (Q3) | Added more experiments and analysis in Appendix A.6 | Pending |
|||2. A detailed comparison with DAPO (Q4)|In the response to Reviewer Kxjd, we provide a detailed explanation from the perspectives of efficiency and capability-boundary expansion.||
|||3. Further analysis on diversity (Q5)|In the response to Reviewer Kxjd, we provide a detailed analysis from the perspectives of generation entropy and pass@k.||
| ugNa | 4 | 1. Misunderstanding of the acceleration mechanism (Q1,Q2,Q6)| In the response to Reviewer ugNa, we provide a detailed clarification. | 6 |
|||2. Lack of ablation studies. (Q3,Q7,Q8)|Added more ablation experiments in Appendix A.5||
|||3. Concerns about the KL coefficient. (Q4,Q10)|In the response to Reviewer ugNa, we provide a detailed explanation and include experimental results for reference.||
| QY8s | 4 | 1. Lack of ablation studies. (Q2)| Added more ablation experiments in Appendix A.5 | Pending |
|  |  | 2. Efficiency Analysis of MENTOR (Q3) | Added more experiments and analysis in Appendix A.6 |
|||3. Concerns about the normalization strategy. (Q10)|We provide a detailed explanation and experimental results in Appendix A.5.4.||
|||4. The effectiveness of MENTOR on strong models. (Q5, Q11)|In the response to Reviewer QY8s, we provide a experiment results on Nemotron-1.5B and include a detailed analysis.||

We are sincerely grateful to all reviewers for their constructive feedback, which has greatly helped improve the quality and completeness of our work and made the technical contributions more solid. We would like to give **special thanks to Reviewers uaif and ugNa** for their positive engagement and recognition of our efforts during the rebuttal period. Finally, we truly value and appreciate all reviewers’ and AC’s contributions to enhancing this work and supporting the research community.

---

### Meta-Review · Area_Chair_jyw9 · 2026-01-05

**Summary:**

The paper proposes a way to improve a model's performance through expert guidance on a subset of the generated tokens. The authors argue that previous methods integrating expert guidance, e.g. performing SFT on expert trajectories, would likely lead to decreased diversity and exploration. To address this, the authors suggest a mixed-policy method (equation 8) which emphasizes some high-entropy tokens in which the expert policy is used to generate tokens. Results show significant improvements across a range of Math-oriented tasks.

The reviewers have raised concerns about the practicality of the method and whether generating tokens from an expert policy would significantly reduce throughput. The authors reported numbers that show that although there is a reduction in throughput, it is still competitve, especially when compared to other expert-guided RL methods. The authors additionally provided a significant number of ablations and comparisons to new baselines which strengthened the claims of the paper.

Overall, the paper proposes a clever way to utilize an expert within RL optimization. However, one of the main claim of the paper is about exploration. There is in fact very little evidence that points to this, as most results simply show increased performance. One could argue that this increase is due to better exploration, but it could just as well be explained as better exploitation thanks the guidance of the expert. Despite this, the paper presents enough results that could be of interest to the community. Future work could investigate in greater detail when the expert guidnace is emphasized and could try to understand better whether entropy-based token selection is the best strategy.

**Reviewer Concerns:**

Reviewer uaif: the authors addressed the concerns about hyperparameters, the strategy for selecting tokens used for expert guidance and the expert's capacity.
Reviewer Kxjd: the authors addressed concerns about the comparison to DAPO and throughput. Concerns about diversity were not fully answered.
Reviewer ugNa: the reviewer raised concerns about about the number of seeds, and removing the KL penalty, which were addressed.
Reviewer QY8s: the reviewer's concerns about ablations and alternate forms of mixing policy were answered in detail. Additional baselines in the form of the Nemotron-1.5B model were also completed. The concerns about a weak experts would likely still raise questions.

**Reviewer Scores:**

Reviewer uaif: It is likely that the score would be increased.
Reviewer Kxjd: it is possible that the score would increase.
Reviewer ugNa: It is likely that the score would be increased.
Reviewer QY8s: It is likely that the score would be increased.

---

### Decision · Program_Chairs · 2026-01-26

Accept (Poster)